# A continuous approach to explain insomnia and subjective-objective sleep discrepancy
Rubén Herzog [1], Flynn Crosbie[2], Anis Aloulou[2,3], Umaer Hanif [2], Mounir Chennaoui[2,4], Damien Léger[2,3,6] & Thomas Andrillon [1,5,6] ✉

Understanding insomnia is crucial for improving its diagnosis and treatment. However, many subjective complaints about insomnia do not align with objective measures of sleep quality, as is the case in subjective-objective sleep discrepancy (SOSD). We address this discrepancy by measuring sleep intrusions and instability in polysomnographic recordings from a large clinical database. Using machine learning, we develop personalized models to infer hypnodensities—a continuous and probabilistic measure of sleep dynamics—, and analyze them via information theory to measure intrusions and instability in a principled way. We find that insomnia with SOSD involves sleep intrusions during intra-sleep wakefulness, while insomnia without SOSD shows wake intrusions during sleep, indicating distinct etiologies. By mapping these metrics to standard sleep features, we provide a continuous and interpretable framework for measuring sleep quality. This approach integrates and values subjective insomnia complaints with physiological data for a more accurate view of sleep quality and its disorders.

Sleeping is essential to our survival[1]. However, between 10 and 20% of adults living in industrialized countries struggle to achieve a good night's sleep and complain of chronic insomnia[2]. Poor sleep and insomnia take a significant toll on general and mental health (e.g., increase in cardiovascular risks, anxiety, depression, accidents, etc)[3–6]. Because of their high prevalence and far-reaching consequences, tracking, managing, and curing sleep difficulties are paramount for public health and quality of life[7,8]. However, the diagnosis and treatment of sleep disturbances can be complex, notably because of a lack of robust biomarkers for sleep quality. This difficulty is partly due to the frequent discrepancy between individuals' subjective reports regarding their sleep and the findings of objective standard sleep exams, a phenomenon referred to as sleep state misperception (SSM) or subjective-objective sleep discrepancy (SOSD)[9,10].

Quantifying sleep is complex and can be done in various ways, such as through subjective reports (e.g., questionnaires, scales, sleep logs, etc), measurements of body activity (e.g., using actigraphy, a wrist-accelerometer), or measurements of brain activity. The latter is performed with the technique of polysomnography (PSG), which consists of the continuous recording of brain (electroencephalography, EEG), ocular (electrooculography, EOG), muscular (electromyography, EMG) activity[11]. PSGs are the gold standard of sleep research and serve as the basis for the definition of sleep and its sub-stages[12,13].

PSGs are required to confirm the diagnosis of all sleep disorders, with the notable exception of insomnia[14,15]. This is perhaps because, in up to 50% of insomnia complaints, the visual inspection of PSG recordings significantly diverges from self-reported estimates of sleep quantity or quality[9]. Consequently, chronic insomnia is defined on subjective reports alone (namely reported difficulties in initiating or maintaining sleep and/or a chronic sensation of non-restorative sleep[16]). However, pharmacological treatments for insomnia still need to be evaluated based on criteria derived from PSG recordings[14]. For example, the American Food and Drug Agency requires positive effects on PSG-derived sleep metrics to authorize new drugs, and the European Medicines Agency recommends using PSGs in clinical trials[17,18].

Beyond the specific case of insomnia, SOSD questions the overlap between the objective and subjective estimates of sleep[10,19,20]. SOSD is present in all sleep disorders[21,22] and, although to a lesser degree, this is a phenomenon also recognized in good sleepers[10] and 20% of hypersomnia complaints. SOSD can be bidirectional and reflect the feeling of being awake while PSG recordings indicate sleep, but also the feeling of being asleep while

[1]Sorbonne Université, Institut du Cerveau - Paris Brain Institute - ICM, Inserm, CNRS, Paris, France. [2]Université Paris Cité, VIFASOM (Vigilance Fatigue Sommeil et Santé publique), Paris, France. [3]APHP, Hôtel-Dieu, Centre du sommeil et de la Vigilance, Paris, France. [4]Institut de recherche biomédicale des armées (IRBA), Brétigny-sur-Orge Paris, France. [5]Monash Centre for Consciousness & Contemplative Studies, Monash University, Melbourne, Australia. [6]These authors jointly supervised for this work: Damien Léger, Thomas Andrillon. ✉e-mail: thomas.andrillon@icm-institute.org

PSG recordings indicate wakefulness[10]. In good sleepers, a recent study stressed the lack of correlation between classical neurophysiological indexes of sleep depth and subjective sleep depth[23]. These results deeply question some fundamentals of sleep science and ask for renewed efforts to identify robust biomarkers of sleep quality in order to better understand what makes a good night's sleep.

Sleep is generally associated with a loss of responsiveness and awareness to the external world, which are accompanied by changes in brain activity with a shift to high-amplitude low-frequency oscillations[20]. Based on this relationship, the current consensus is that the detection of certain EEG hallmarks (e.g. slow waves, sleep spindles) is enough to identify the state of sleep and to infer sleep depth. However, recent discoveries show that this characterisation of sleep and sleep depth is too coarse[23], does not capture fine transitions at sleep onset or between sleep sub-stages[24], and most importantly does not always align with subjective feelings[25]. This could be first because standard PSGs rely on a small number of EEG electrodes (typically just 3), providing a partial reading of sleep state and its local regulation[26,27]. This is important because local modulations of sleep depth can account for the feeling of being awake while asleep[23], the occurrence of dreaming[28], the sensitivity to sensory inputs during sleep[29], and sleep disturbances such as insomnia[30]. Second, relying on the visual inspection of sleep recordings could mask important signs of sleep disturbances that are not visible to the naked eye[31,32]. For example, fast oscillations are difficult to track visually, and are even filtered out when specialists inspect PSGs, despite the fact that several studies have linked these fast oscillations to SOSD[10,31,33]. A recent examination of the topography of PSG has associated diffuse cortical hyperactivations with SOSD, identifying a decrease in the delta/beta ratio as a major correlate[34]. However, for typical lower-density clinical PSG recordings, it is still difficult to track the subtleties that could explain how a subjective complaint of insomnia can coexist with what appears to be, at the surface, normal sleep on the PSG[31].

New methods in sleep medicine could help reduce the gap between objective and subjective assessment of sleep and uncover the root causes of poor sleep quality. With increasing computational power, it is now easy to extract hundreds or thousands of different features even in large datasets[35], considerably enriching the quantity of information derived from a single PSG recording. This approach corroborates and nuances the standard classification of sleep in a data-driven way[32]. Also, advances in machine learning further enhance this potential for identifying the optimal set of features for classifying a specific sleep disorder, uncovering new biomarkers for diagnosis and elucidating new mechanisms for more targeted and effective treatments. Furthermore, recent studies on automated sleep stage classification illustrate these opportunities[36]. Visual scoring of sleep stages, as rapid eye movement (REM) or non-REM (NREM) sleep, crucial yet time-consuming, achieves around 80% interscorer agreement[37], while machine-learning techniques are capable of reaching human performance levels in healthy and clinical groups[38–40]. Beyond practical classification, these algorithms offer a probabilistic perspective on sleep by computing the probabilities of the five standard sleep stages, known as hypnodensities, which provide a finer and continuous description of sleep, that could aid in diagnosis[38–40].

Here, we set out to leverage innovations in big data and machine learning for the detection of insomnia and SOSD. We retrospectively analyzed a large cohort of patients diagnosed with chronic insomnia based on subjective complaints[16]. We sought to identify the neural signatures that distinguish between insomnia with and without SOSD using PSG recordings. While prior research has highlighted differences in various markers of sleep quantity and quality[10], accurately identifying insomnia with SOSD based solely on PSG data remains a challenging endeavor. All these patients underwent a PSG and we found evidence for a significant SOSD in a subset of these patients (SOSD+ group, 24% of total) and no evidence of a significant SOSD in the rest (SOSD− group) (see[31] and Methods for details) although the exact definition of what constitutes SOSD varies from one study to another[41]. We compared these two insomnia groups with healthy volunteers (good sleepers, the GS group). From PSG recordings, we

extracted a set of highly discriminative and minimally redundant features[42], which we used to train and validate a personalized model of sleep architecture. From this personalized model, we extracted hypnodensities to train a cross-subject algorithm for the detection of insomnia with or without SOSD, reaching an overall accuracy of 0.77 ± 0.017%. We finally interpreted the internal functioning of the algorithm (i.e., opening the black box) to identify the neurophysiological signatures of insomnia and revealed evidence for sleep/wake mixing in both sleep and wakefulness for the SOSD− and SOSD+ groups. The SOSD+ group was further characterized by higher probabilities of sleep within wakefulness and higher wake instability, suggesting that SOSD stems from a perturbation of both sleep and wakefulness. Finally, our approach was also sensitive to severity of objective impairment of polysomnography in insomnia as we could accurately predict markers of sleep quality (total sleep time [TST], sleep onset latency [SOL], wake after sleep onset [WASO] and sleep efficiency [SE]) from PSG recordings. Overall, this work shows that finer analyses of PSG recordings can close the gap between subjective complaints of insomnia and objective quantifications of sleep quality.

## Results
### Hypnodensities: leveraging a probabilistic approach to sleep staging

We examined a large database of PSG recordings ($n = 927$) to identify sleep quality indicators that vary between the three groups of sleepers: Good sleepers (GS, $n = 104$), patients with insomnia without sleep state misperception (SOSD−, $n = 624$) and patients with sleep misperception (SOSD+, $n = 199$) (Fig. 1A). The clinical dataset utilized was diverse in terms of recording devices, electrode layout, and sampling rates, making direct comparisons challenging. To overcome this issue, we developed an analytical framework that leverages the estimation of hypnodensities (HD)[38]. Expanding on the standard discrete sleep staging, HD provides a nuanced assessment of the probability of each sleep stage (wake, N1, N2, N3, and REM) within a given epoch. From the HD computed on each epoch, we measured the level of stage mixing and epoch instability (see *Methods: Intrusions and instability*), and used these measures as biomarkers to predict the clinical group and standard sleep quality metrics at the individual level (Fig. 1B). Sleep and wake intrusions were defined respectively as the probability of wakefulness during sleep epochs and the probability of sleep during wake epochs.

We first devised a computationally-light approach to compute HD from PSG recordings. We employed *catch22*[42], a robust feature extraction algorithm, to extract features from each EEG and EOG channel, enabling us to project each 30-second epoch into a multidimensional space. Then, by leveraging the sleep scoring done by experts (see *Methods: Hypnograms*), we trained a machine learning (ML) algorithm to predict the hypnogram based on these features, yielding a HD for each epoch (see *Methods: Data harmonization by hypnodensity estimation*). Finally, we used tools from information theory, namely the entropy and the Kullback-Leibler divergence ($D_{KL}$) to quantify the level of intrusion and instability of each HD, respectively (Fig. 1C; see *Methods: Intrusions and instability* for a detailed definition and interpretation of these metrics).

### Increased intrusions in insomnia: sleep intrusion in SOSD+ and wake intrusion in SOSD−

We started by comparing our predicted sleep staging with the sleep staging obtained with human sleep experts and selecting only subjects where our pipeline yielded hypnogram prediction accuracy larger than 0.4 (Supplementary Fig. 1A, B. 115 subjects were discarded, whose data was not included in this work). As a first sanity check, we confirmed that, on average, the maximum of each HD matched its corresponding epoch stage (Fig. 2A), with the exception of N1, which was also comparatively poorly predicted by the algorithm (Supplementary Fig. 1C, D). For this reason, N1 was excluded from the expert stages to be analyzed, but it was nevertheless considered in the computation of HD, entropy and DKL for completeness. In the

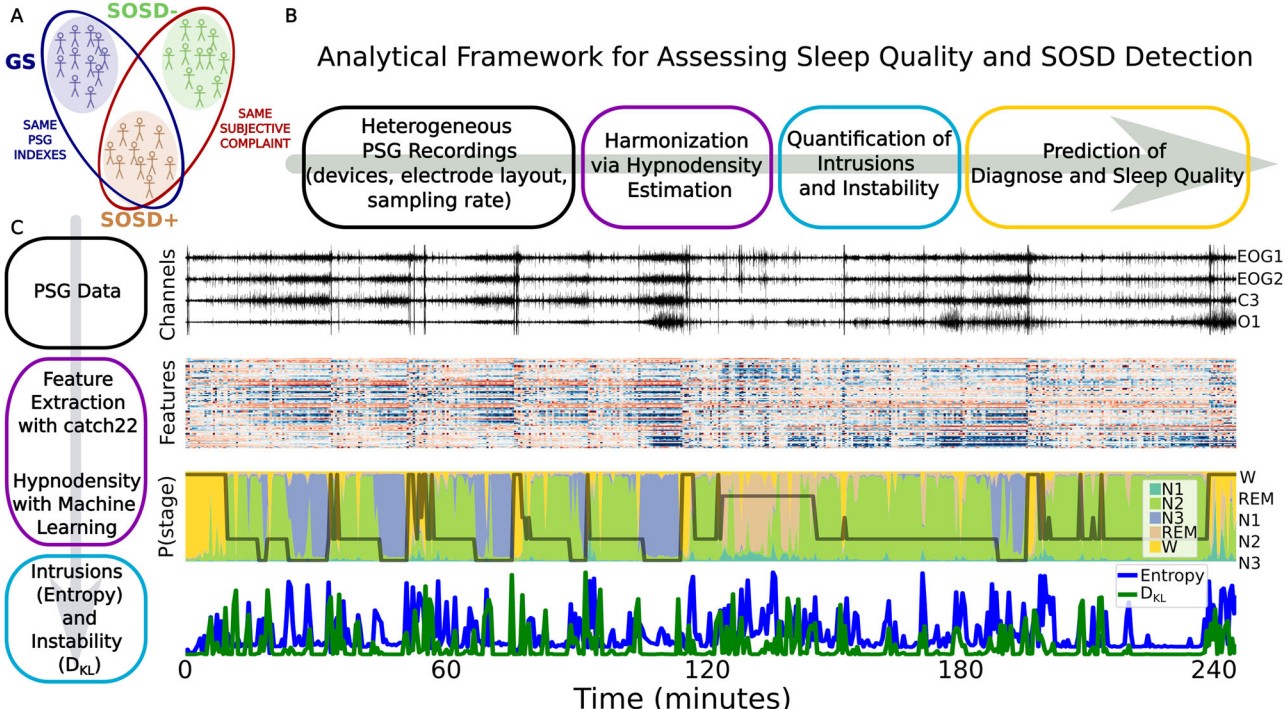

**Fig. 1 | Analytical pipeline for characterizing sleep quality and addressing subjective-objective sleep discrepancy (SOSD). A** We analyzed PSG data from a population of Good Sleepers (GS), patients with insomnia without SOSD (SOSD−) and with SOSD (SOSD+ ). **B** Heterogeneous PSG recordings were harmonized via the estimation of hypnodensities, which enabled the quantification of the level of intrusion and instability of sleep dynamics. Finally, intrusions and instability were used as markers for predicting individual diagnoses and sleep quality measures. **C** PSG recordings (example of the first 4 h of sleep of a GS subject) were scored by experts in 30 s epochs (hypnogram) and then a set of features (*catch22*) were extracted to characterize each epoch in a multidimensional space. A machine learning algorithm was trained to predict the hypnogram. This also yielded, for each epoch, the probability of each stage, i.e., the hypnodensity (colors). Finally, the hypnodensities were analyzed with tools from information theory to measure the level of intrusions (entropy) and instability ($D_{KL}$) of each epoch.

**Fig. 2 | Intrusions and instability on good sleepers and in insomnia with and without SOSD.**
**A** Boxplots with average hypnodensities inferred for each sleep stage scored by experts and for each sleep group (colors). Black horizontal lines in the top of the panels denote significant differences between groups (* represents Bonferroni corrected Kruskal-Wallis $p < 0.001$). **B**, **C** The respective average level of intrusion (entropy) and instability ($D_{KL}$) for each sleep stage scored by experts and sleep group, respectively. Black horizontal lines on top of the panels denote significant pairwise differences (+ represents Bonferroni corrected Wilcoxon rank sums $p < 0.001$). SOSD−: $n = 624$, SOSD + : $n = 199$ and GS: $n = 104$; Each sample corresponds to a different participant.

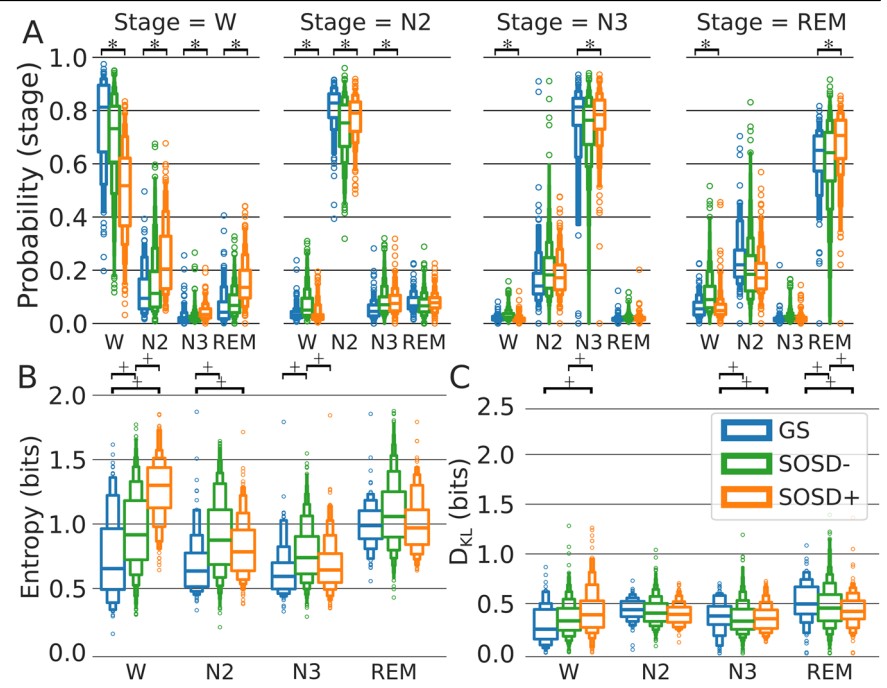

following, any mention of sleep stages will refer to those scored by experts, while mentions to stage probabilities or intrusions refer to the probabilities obtained from the hypnodensities.

When examining group differences, we found a significant reduction (Bonferroni corrected Wilcoxon-rank sum $p < 0.001$) of the wake probability during expert-scored wakefulness for SOSD− (0.70 ± 0.16) and even less for SOSD+ (0.49 ± 0.17), compared to GS (0.75 ± 0.18) (see Supplementary Table 1 for HD details). While SOSD− exhibited a significantly larger probability of wakefulness during sleep (NREM + REM), SOSD+ did not (GS: 0.091 ± 0.14; SOSD−: 0.12 ± 0.12; SOSD + : 0.067 ± 0.080); Bonferroni corrected Wilcoxon-rank sum GS vs SOSD− $p < 0.001$, GS vs SOSD+ $p > 0.01$). In fact, SOSD− exhibited larger wake intrusions (defined here as the wake probability in sleep epochs) than SOSD+ in all sleep stages. Thus, our results show that both SOSD− and SOSD+ insomnia sub-types are associated with sleep intrusions during wakefulness, but in the case of SOSD+ sleep intrusions are not accompanied by wake intrusions during sleep. These results are in line with the apparent normal sleep characteristic of SOSD+ (see Supplementary Table 2–5 for details on sleep microstructure of the groups).

To quantify and compare the levels of intrusion between sleep group and sleep stages, we computed the entropy of each HD (see *Methods: Intrusions and instability*). Entropy is a function of the full HD that measures the proximity to a uniform distribution, thus providing more information than just measuring the highest probability on the HD. As high HD entropy indicates a high level of intrusion (proximity to a uniform distribution), high entropy during wakefulness implies sleep intrusions, while high entropy during sleep implies wakefulness or intrusions from other sleep stages. Consistent with the previous results, we found a significant increase in HD entropy during wakefulness in both insomnia groups (with and without SOSD) compared to GS and also in SOSD+ compared to SOSD− (Fig. 2B, GS: 0.74 ± 0.34 bits; SOSD−: 0.95 ± 0.30 bits; SOSD+ : 1.27 ± 0.24 bits; Bonferroni corrected Wilcoxon ranksum p < 0.001 for all pairwise comparisons). Note that these differences held even after controlling by the stage normalized frequency (i.e., the number of epochs of each stage divided by the total number of epochs; see Supplementary Table 6). Also, SOSD− had a significantly larger entropy during N2 and N3 than GS, and a significantly larger entropy in N3 than SOSD+ (Bonferroni corrected Wilcoxon ranksum $p < 0.001$). This increase in entropy in NREM sleep in the SOSD− group can be largely attributed to an increase in wake probabilities. In contrast, while SOSD+ also has a significantly larger entropy in N2 than GS, this increase was not related to an increase in wake intrusions, but rather to an increase in the probability of N3 (so deeper sleep). See Supplementary Table 7 for details.

To further characterize the sleep dynamics, we quantified the level of instability between two consecutive HDs within the same sleep stage by the $D_{KL}$. A low $D_{KL}$ means that both epochs are very similar in terms of their HDs (i.e. high epoch-to-epoch stability), while a high value means that two consecutive epochs largely differ. Note that this is different from just computing the difference of entropies, as two HDs could be different but still have the same entropy (e.g. same probabilities but assigned to different stages). We found that SOSD+ had larger wake instability than GS and SOSD− (Fig. 2C, Bonferroni corrected Wilcoxon ranksum $p < 0.001$. These differences held after controlling for sleep stage normalized frequency), while SOSD− did not significantly differ from GS. No differences were found for N2. Both SOSD− and SOSD+ had less instability than GS for N3, with no significant differences between them. Finally, REM was more unstable for GS than in SOSD− and SOSD+ , with SOSD+ exhibiting less instability than SOSD−. However, after controlling by the stage normalized frequency, the difference between SOSD+ and SOSD− on REM vanished (see Supplementary Table 8). To summarize, reduced instability in N3 and REM is a common factor in the perception of poor sleep quality in SOSD− and SOSD+ . Yet, only SOSD+ shows an increased instability in wakefulness compared to GS. Again, these results are consistent with the condition of SOSD+ as a disruption of intra-sleep wakefulness rather than sleep itself.

## Sleep intrusions and wake instability predict diagnosis and sleep quality

To further confirm that intrusions and instability were reliable markers of sleep groups, we used these features as predictors in a machine learning classifier (XGBoost, see *Methods: Matching learning classifier and regression*). We performed two different types of classifications: a 3-class (GS vs SOSD− vs SOSD+ ) classification and a within-insomnia 2-class (SOSD− vs. SOSD+ ) classification. In detail, we calculated the average and standard deviation of intrusions and instability for each sleep stage, yielding a total of 16 features (4 features per sleep stage, 4 sleep stages analyzed). We then applied feature selection to select the smallest set of features that maximizes classification performance. First, we sequentially included the features in the classifier according to their maximum relevance minimum redundancy score (MRMR, see *Methods: Feature selection*). To avoid biases given by size imbalances, we took 30 random subsamples of a size equivalent to the the smallest sample size (GS in the 3 class problem and SOSD+ in the binary problem, see *Methods: Matching samples by size, sex, age and BMI*) and ran a repeated 5-fold cross-validated classifier. In both cases, the 3 first features were intrusions in wakefulness, instability in wakefulness and the standard deviation of instability in REM. We found better performance for the 2-class than for the 3-class classifiers (Fig. 3A), but, in both, the performance was saturated with less than 10 features. We examined the confusion matrices (Fig. 3B, C) associated with the points before performance saturation (vertical lines in Fig. 3A; 9 and 4 features for the 3-class and 2-class classifiers, with average AUC of 0.82 ± 0.016 and 0.86 ± 0.013, respectively) and confirmed a strong diagonal in both cases. We repeated the analysis for the 2-class classifier using the best 4 features in subsamples matched for age, sex and body-mass index and found a small performance decrease (Supplementary Fig. 2, average AUC of 0.83 ± 0.014 and 0.86 ± 0.013 for matched and unmatched samples, respectively; Wilcoxon ranksum $p < 0.001$). Finally, we repeated the analysis using hypnodensities derived from the U-Sleep algorithm[40], which is a state-of-the-art algorithm for automated sleep staging and hypnodensities estimation. This additional analysis was also designed to test if an algorithm trained on a different dataset would extract hypnodensities that allow the classification of the three groups even when the model is not trained on the PSG recordings themselves. Following the same feature selection and inclusion procedure, the hypnodensity derived from this external model yielded lower but above chance classification performance (Supplementary Fig. 3). These results are consistent with the previous section and demonstrate that intrusions and instability in wakefulness not only characterize the differences between sleep groups, but they can also be used to distinguish insomnia subtypes at the individual level.

However, the classifiers did not reach a perfect performance and to better understand cases of misclassification, we related the subject prediction accuracy of the 2-class classifier with the sleep metrics used for diagnosis (Fig. 3D, E; see *Methods: Participants and diagnose*). Subject prediction accuracy was obtained by running 1000 iterations of the repeated cross-validated classifier and counting how many times a subject was correctly classified in the test sample. This value is 1 when the subject is correctly classified in all the test samples and 0 for the opposite case. We found that most of the misclassifications occur for patients close to the cut-offs used for diagnosis (see *Methods: Objective insomnia*), suggesting that the thresholds used for sub-typing of insomnia could be hiding a continuum within insomnia.

To further explore the idea that intrusions and instability could index sleep quality in a continuous way, we used the same features to predict the values of the sleep quality metrics in insomnia patients. TST, SOL, WASO and SE are critical metrics for the diagnosis and evaluation of insomnia severity[16]. To do so, we used 120 iterations of a repeated 5-fold cross-validated ML regression algorithm. As for the previous classification problem, we used the MRMR scores to include features sequentially as predictors of the sleep metrics and measured the regression performance using the RMSE (Fig. 4A). In all cases we found that adding more features improved the performance, with no evident saturation point this time. All

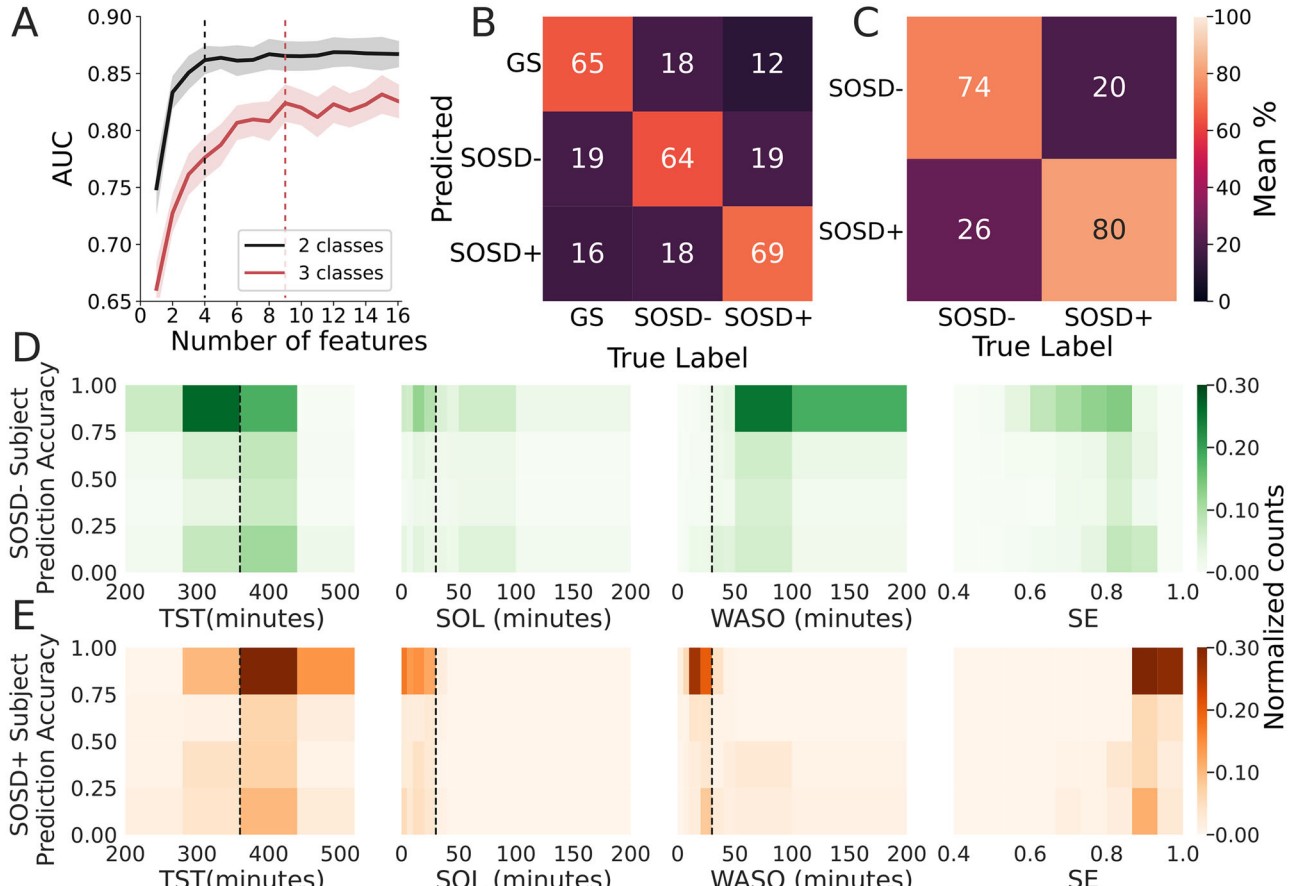

**Fig. 3 | Sleep intrusions and wake instability predict sleep groups. A** Classifier average performance as a function of the number of features included as predictors for binary (black, only SOSD- and SOSD+ ) and 3 class problems (red, GS, SOSD- and SOSD+ ) using size-balanced subsamples of data. For the 3-class problem all random subsamples of SOSD− and SOSD+ comprised 104 subjects, while for the binary problem random SOSD− subsamples of 199 subjects were taken, so all the sleep groups had the same sample size. Solid line and shaded area correspond to the average and 1 standard deviation, respectively, of 60 iterations of a repeated 5-fold cross-validated classifier, using 20 random subsamples each time. **B**, **C** Average confusion matrix for the 3 and 2-class problem, corresponding to the vertical lines in (**A**). **D**, **E** 2D Histograms of the class prediction accuracy for each subject in the binary problem and sleep quality metrics for SOSD− and SOSD+ , respectively. Color scale represents the percentage of the population and vertical dashed line represents the cut-off for the diagnosis associated with each sleep metric.

the predictions were significantly correlated with the veridical variables obtained from PSG recordings (Fig. 4B). We obtained the best correlations for WASO followed by SE, TST, and SOL. Predictions were also better for SOSD− than for SOSD+ . The average intrusions and instability in wakefulness had the largest correlation with WASO and SE (Fig. 4C), with a negative correlation for the former and a positive for the latter. Accordingly, our results suggest that the level of intrusions and instability in wakefulness can not only discriminate between sleep groups, but also predict the severity of objective markers for the diagnosis of insomnia.

**Sleep intrusions correlate with low-frequency EEG activity**

Finally, we aimed to understand intrusions and instability in terms of PSG macro and microstructural features (transitions between sleep stages and spectral analysis, respectively). Following previous results on dissociated states in sleep and wakefulness[20,26] in particular in insomnia and SOSD[23,30], we expected that sleep intrusions during wakefulness would be characterized by increased EEG power in low frequencies, while for intrusions during sleep we expected an increase in the power of high frequencies. Accordingly, we found that intrusions in wakefulness were positively correlated with the power in delta and theta bands, while for sleep stages intrusions were positively correlated with the power in high frequencies, specifically in the beta and gamma bands (Fig. 5A). This effect was present regardless of the EEG electrode position (Supplementary Fig. 4). Notably, in N3, intrusions were inversely correlated with the power in the delta band, which is often

used as a proxy for sleep depth[43]. The same trend was observed when splitting by sleep groups (Fig. 5D), with subtle differences within each sleep stage and frequency band. For example, intrusions for SOSD− and SOSD+ were more correlated than GS for low-frequency bands (delta) in wake and for high-frequency bands (gamma) in N3. Note that our procedure to infer HD did not involve measuring power in these bands, demonstrating that despite the novelty of our approach it is highly consistent with previous results on dissociated sleep states[10,44,45].

Instability followed a similar but lesser tendency when correlated with power in frequency bands (Fig. 5B). A more detailed analysis revealed that instability within sleep stages correlated more with increases in the power of high frequencies in two consecutive epochs (Fig. 5C), especially in N3, which was also present when splitting by sleep groups (Fig. 5E). Moreover, instability in wakefulness was negatively and significantly correlated (Bonferroni corrected Pearson $p < 0.001$) with the probability of wake self-transition, an observation that held when splitting by sleep groups (Fig. 5F).

In conclusion, our results show that insomnia and insomnia subtypes can be detected by an automated analysis of hypnodensities. Furthermore, our results show evidence of wake intrusions during sleep for SOSD− patients and of sleep intrusions during wake both SOSD− and SOSD+ patients. This is consistent with previous findings showing that sleep intrusions during wakefulness are associated with increases in EEG low frequency power[20] and that wake intrusions during sleep are associated with increases in high-frequency EEG power. Thus, our approach does not only

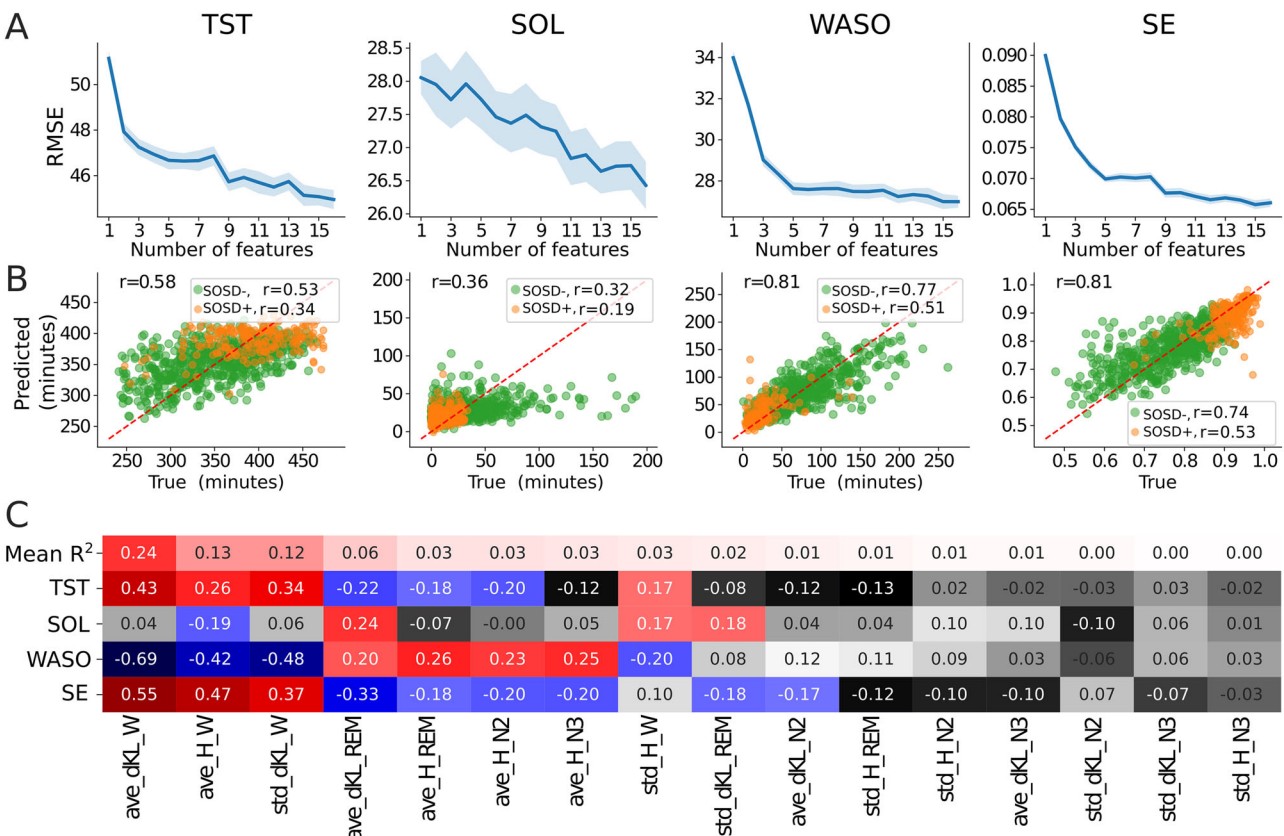

**Fig. 4 | Intrusions and instability predict sleep quality. A** Root mean squared error (RMSE) for a ML regression as a function of the number of features included as predictors for each of the 4 sleep parameters, using only SOSD+ and SOSD−. Lines and shaded areas are the average and standard deviation, respectively, out of 120 iterations of a repeated 5-fold cross-validated ML regression algorithm. **B** Average predicted vs true sleep parameter values corresponding to using all the 16 features. Subject averages were obtained from the predictions in the test samples. Colors denote sleep groups and the diagonal dashed red line is the identity line. All metrics significantly correlated with their prediction ($p < 0.001$), and the correlation followed WASO > SE > TST > SOL. Predictions were better for SOSD− than for SOSD+. **C** Spearman correlation matrix between the features and the sleep quality parameters. Only colored squares were significant (Bonferroni corrected $p$-value < 0.001). Features are sorted in descending order following the average $R^2$ (first row of the matrix). Features are encoded as 'ave' for average, 'H' and 'dKL' for entropy and $D_{KL}$, respectively, and the last word corresponds to the sleep stage. For example, ave_H_W is the average entropy of wakefulness. Note that average intrusions and instability in wakefulness show the largest average $R^2$ and the largest correlations with WASO and SE.

lead to good classification performance but also allows us to interpret the underlying features, shedding a new light on the physiopathology of insomnia.

## Discussion

We aimed to identify neural signatures differentiating insomnia with and without SOSD using PSG recordings. Previous research demonstrated differences in various markers of sleep quantity and quality but the identification of insomnia with SOSD on PSG alone proved difficult[10], which potentially stresses the need for large sample sizes and finer methods. In particular, PSG datasets can include large differences both within and between datasets, such as variations in the equipment used, the number or location of EEG electrodes and the recording or preprocessing routines. We addressed this issue by implementing data harmonization through the extraction of hypnodensities. Our aim here was to mitigate dataset-specific idiosyncrasies and enhance comparability across studies and experimental setups. From a more fundamental perspective, hypnodensities also capture the fact that sleep is not composed of discrete, mutually exclusive states[20,24] but mixed sleep states are both frequent[46–48] and informative of sleep quality[23,31]. By leveraging information theory and ML techniques to analyze hypnodensities, we provided a nuanced understanding of insomnia, potentially applicable to other sleep disorders[49]. Our findings revealed significant differences in the physiological profile of SOSD+ and SOSD−, despite similar subjective complaints, reinforcing the view of SOSD+ as a distinct insomnia sub-type[50]. Both conditions exhibited increased sleep

intrusions during intra-sleep wakefulness, with a higher magnitude observed in SOSD+. In contrast, only SOSD− was characterized by significant wake intrusions during sleep. This distinction aligns with the "era of intrusions" described by Stephan and Siclari and give support to the hypothesis that SOSD is actually a mismeasurement rather than a misperception[10]. Despite the lack of spatial resolution of our work, our results are compatible with the recent finding of diffuse cortical hyperarousal as a marker for SOSD[34], which could be also interpreted as sleep/wake mixing. Our results emphasize the necessity to shift from binary diagnoses (as SOSD− vs SOSD+) to a graded diagnostic framework that better embraces the variability within sleep and between individuals[24].

Our results highlight the contribution of both sleep and wakefulness to insomnia, in line with the hyperarousal model of insomnia[51]. According to this model, insomnia would result from heightened levels of physiological and psychological arousal throughout the day and night. In support of the hyperarousal model, individuals with insomnia show increased cardiac activity both before and during sleep, elevated temperature during sleep, increased metabolism during wakefulness, and increased levels of cortisol during wakefulness (see refs. 52,53 for reviews). In our analyses, we found markers of wake intrusions during sleep for the SOSD− group (higher entropy and wake probabilities) but not for the SOSD+ group (Fig. 2), while intrusions and instability in wakefulness were good predictors of some of the commonly used sleep quality metrics (SE, TST and WASO). Correlations with spectral features indicate that this higher entropy is positively correlated with increase in fast wake-like frequencies (beta, gamma; Fig. 5). We

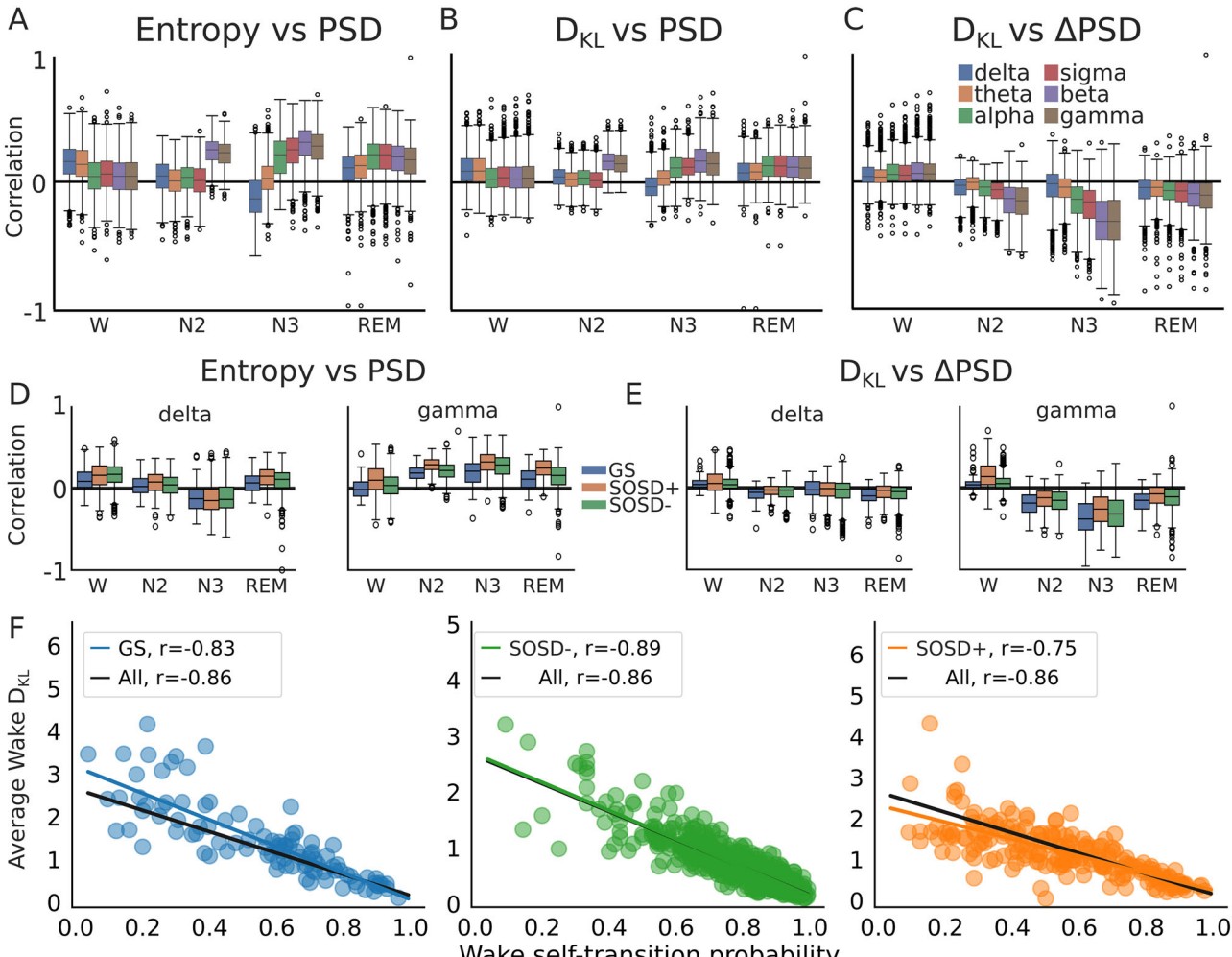

**Fig. 5 | Interpretation of intrusions and instability in terms of macro and microstructural PSG features. A** Boxplots with the population correlations between entropy and spectral power in 6 EEG frequency bands for electrode C3. **B** Same as (**A**), but for the $D_{KL}$. **C** Same as (**B**), but using the difference between power in two consecutive epochs within the same sleep stage. $\Delta$PSD corresponds to PSD(t) - PSD(t + 1), where t corresponds to a specific epoch. **D** Same as (**A**), but splitted by sleep groups, and showing only delta and gamma bands. **E** Same as (**D**), but for the $D_{KL}$ vs $\Delta$PSD. **F** Correlation between the average $D_{KL}$ in wakefulness and the wake self-transition probability, for the 3 sleep groups (colors). Black line is the correlation for the whole population and colored lines are the correlation for each sleep group.

thus found evidence of hyperarousal during sleep but only for SOSD− patients. SOSD+ patients showed a different pattern of results with an increase in entropy and Kullback-Leibler divergence, depicting intrusions and instability respectively, mostly during wakefulness (Fig. 3). Intrusions during wakefulness were characterized by increased low-frequency EEG activity (Fig. 5). As in the SOSD+ group, SOSD− patients also showed intrusions during NREM sleep, this time correlating with fast-frequency EEG activity. First, these results suggest that the hyperarousal model may not fully apply to SOSD+, for which wakefulness shows signs of hypo-aoausal rather than hyperarousal. Second, SOSD− and SOSD+ could represent different extremes of disrupted arousability leading to a subjective feeling of a lack of sleep: the former by hindering the ability to fall asleep and the latter by compromising wakefulness quality. This novel insight could not only inform diagnosis but also treatments for insomnia based on these different processes, potentially shifting the clinical management of SOSD+ from pharmacological treatments seeking to improve sleep to therapeutic approaches seeking to improve daytime vigilance such as light therapy or physical exercise[15]. Indeed, the current reference treatment for insomnia is cognitive-behavioral therapy for insomnia (CBTi), whose effectiveness is recognized regardless of the type of chronic insomnia[51]. We propose that the notions of "intrusions", "stability" and "mixing" of sleep states represent

useful concepts to explain the correspondence between what is measured by the PSG and what is perceived by individuals, to enable certain patients to better understand the complexity of brain states during sleep. These notions could help practitioners explain to patients the discrepancy between a normal PSG outcome and their insomnia complaint. Prospective studies could also leverage this new approach to personalise treatments by investigating if metrics derived from PSG recordings can predict treatment outcomes.

We confirm our previous findings[31] that insomnia can be detected from PSG recordings with high performance (AUC: 0.86; Fig. 3) and extend them by showing that the same PSG recordings can be used for the sub-phenotyping of insomnia and the distinction between SOSD+ and SOSD− (AUC: 0.82; Fig. 3). Furthermore, we highlight the limitations of traditional discrete classifications of SOSD− and SOSD+, suggesting that a continuum model more accurately captures the complexities of these conditions at the individual level. We were indeed able to identify markers that provide information about sleep quality (SE, TST, WASO, etc; Fig. 4) beyond binary classifications, potentially capturing the severity of objective markers of insomnia and offering a PSG inference of sleep quality. Moreover, even when our classifier made errors, we observed that these errors occurred for patients near the diagnosis cut-offs, indicating that individuals who are close

to these hard boundaries are logically more often misclassified (Fig. 4). Based on the new insights, we propose to move towards a continuous and multi-dimensional model of insomnia for which SOSD could represent one of the dimensions. This continuous approach could bypass the complex and unresolved issue of a common definition of what constitutes SOSD[41]. Our hypnodensity-based approach could be further leveraged to find markers specific of different dimensions of insomnia (SOSD, physiological arousal, psychological arousal, chronotype, etc). Our analytic pipeline being light and fully automated, it is both accessible and scalable and could be routinely deployed in sleep clinics. In the future, it could also be deployed to hypnograms derived from other recordings than PSG recordings[54]. Finally, a continuous and multi-modal approach enriched by features extracted from PSG recordings could help implement a precision medicine for insomnia[55].

Our analysis, though utilizing high-level metrics such as entropy and Kullback-Leibler divergence, could be effectively mapped to traditional and well-established EEG spectral features, and also to sleep metrics as SE, WASO and TST. This mapping not only validates our approach but also enhances its interpretability. For example, low-frequency EEG activity characterizes intrusions during wakefulness (i.e., high wake entropy), while high-frequency activity marks intrusions during sleep (high sleep entropy). Also, increases in high-frequency activity between consecutive epochs during deep sleep correlated with higher instability (high Kullback-Leibler divergence). These findings resonate with previous findings on local modulations of sleep and wakefulness[20,26]. Additionally, our intrusions and instability features relate to dynamical macrostructural PSG features like self-transitions between wakefulness (Fig. 5D), which are also disrupted in insomnia[31,56]. Unlike previous work that used black box deep learning models[38,40], our algorithm is fairly simple and can be explainable. This transparency facilitates clinical application, as it makes our findings easier for clinicians to interpret and utilize, and bridges the gap between advanced computational metrics and traditional sleep markers. Potential optimization could provide an estimate of the susceptibility of a given patient to psychological interventions as cognitive behavioral therapy for insomnia[57].

Previous findings on sleep and SOSD could explain the increase in entropy and instability that we observed in the two insomnia groups (Fig. 2). Indeed, an increase in faster oscillations is associated with the feeling of being awake while asleep (FAWS), as assessed directly when awakening individuals from sleep[23]. In insomnia patients, this association between FAWS and faster oscillations is present in both NREM and REM sleep[23]. It is likely that such increases in faster oscillations within sleep epochs would lead to an increase in the entropy of hypnodensities similar to the one we report here. Such modulations of faster oscillations within sleep have also been associated with parasomnia[58], dreaming[28], and cognitive processes during sleep[59], which suggests links between local modulations of sleep and a broad range of sleep disruptions including insomnia with SOSD[26]. Local modulations of cortical dynamics could thus help closing the gap between the objective and subjective assessment of sleep by stressing the importance of global and local dynamics within sleep[20,24]. Our approach could help quantify these local changes beyond the standard global classification of sleep stages. These local dynamics could themselves be caused by a misalignment of sleep with the circadian phase, metabolic changes (e.g., higher metabolic activity during sleep in certain brain regions), or a disruption of the neuromodulation of sleep/wake transitions (e.g., a higher noradrenergic activity during sleep)[20]. These factors have been shown to impact sleep[60-63] and provide interesting avenues of research to better understand and treat insomnia[10,51].

While our study provides significant insights into insomnia, several limitations should be acknowledged. First, this is a mono-centric retrospective study although we controlled for sex, age, size, and BMI but groups were not matched for these variables. We could not account however for all comorbidities, which are common in insomnia[64]. Nonetheless, this dataset is close to real clinical data recorded in sleep clinics, which eases its generalisability when translating these results into clinical applications. Additionally, our study lacked controlled wake recordings, preventing us from

assessing the level of sleep intrusions during normal daytime wakefulness outside of the period just before and during sleep. This limitation could explain our poor prediction of SOL, as we lacked information about pre-sleep activity. However, the strong predictive power and interpretability in terms of traditional EEG features suggest that our procedure is actually capturing distinct underlying physiological processes. In fact, when using an alternative automated sleep scoring algorithm (U-sleep, not trained with insomnia data), we found a bad performance, stressing the fact that our dedicated specific algorithm captures physiological relevant features of insomnia and sleep quality. Despite these limitations, our findings offer intriguing avenues for future research, particularly in exploring both hyperarousal and hypoarousal more comprehensively. Another limitation is that the differences we observe between SOSD− and SOSD+ relate to a single night of polysomnography per subject, which is compared to a subjective complaint of insomnia that covers a previous and longer period of time. This is in line with the data that can be available to practitioners (a subjective and chronic complaint of insomnia associated with a punctual assessment of sleep through a PSG). Our stance also corresponds to the classical definition of SOSD in clinical practice (e.g., the definition of "paradoxical insomnia"[65]). While this approach is well-suited for clinical practice and the identification of biomarkers of SOSD, it is limited in its ability to uncover the underlying neural mechanisms of SOSD. To address the question of mechanisms more specifically, obtaining subjective assessments of sleep, particularly following spontaneous or induced awakenings, is preferable[23]. In addition, our approach does not account for the night-to-night variability in PSG recordings, and phenomena such as the first-night or reverse first-night effects[66,67]. It is thus possible that the SOSD+ and SOSD− labels could change from night to night at the individual level. However, our goal was to characterise the phenomenon of SOSD in general, not to draw detailed inferences on specific individuals. Future studies could employ other approaches to characterise SOSD for a specific night by comparing PSG recordings with subjective assessment of the same night. The same analytic approach as presented here could be applied.

Finally, we propose redefining the terminology used to SOSD in the scientific literature and in the information provided to patients. Terms such as "subjective insomnia", "paradoxical insomnia" and SSM do not reflect the fact that patients with or without SOSD both follow the standard criteria for the diagnosis of insomnia. These terms also imply that the discrepancy is due to a problem of incorrect sleep perception and not an incorrect sleep quantification[10]. Terms such as "intrusions", "instability" and "mixing" of sleep states can offer a more accurate description of sleep, its complexities and disorders[68-70]. By correlating the subjective complaint with physiological data, we aim to contribute in bridging the gap between subjective reports and objective measurements. We favor the idea that SOSD is actually a mismeasurement rather than a misperception[10], and that is a core dimension of insomnia (and sleep in general) rather than a sub-type of insomnia[50]. Our work illustrates how subjective experience is a fundamental dimension of sleep[20], and taking it into account enables a better explanation of the inter-individual variability observed in neural activity. This integrative approach could provide a more graded and comprehensive understanding of sleep and consciousness research in general, enhancing both clinical and research perspectives.

## Methods
### Participants
The cohort examined in this study is an extension of a previously published study[31]. The total number of available recordings was 1042. However, 115 recordings were discarded because of noisy data or because the hypnogram was predicted with a balanced accuracy lower than 0.4 (see below). These recordings were excluded from all analyses, leading to a total sample size of 927 recordings. These excluded files are not included in the reported sample sizes or results. The remaining 927 participants had an age distribution of 44.4 ± 14.0 years and were 66.3% of females (Table 1 and Supplementary Table 9). Each participant underwent a single night of PSG to assess their sleep patterns. The recruitment strategy differed between patients with

**Table 1 | Participants characteristics**

| Variables | | SOSD− (*n* = 624) | SOSD+ (*n* = 199) | GS (*n* = 104) |
|---|---|---|---|---|
| Anthropometric data | Sex (% female) | 69.7 (*n* = 435) | 74.4 (*n* = 148) | 30.8 (*n* = 32) |
| | Age (years) | 46.8 ± 14.1 | 40.8 ± 12.1 | 36.2 ± 11.9 |
| | BMI (kg/m$^2$) | 23.1 ± 3.86 | 23.0 ± 3.56 | 24.2 ± 2.61 |
| Comorbidities | Smoking (%) | 21.8 | 23.6 | 27.3 |
| | Depression (%) | 25.0 | 22.0 | 0.0 |
| | Anxiety (%) | 19.0 | 19.4 | 0.0 |
| | Cardiovascular disease (%) | 13.8 | 5.2 | 0.0 |
| | Pulmonary disease (%) | 5.2 | 5.2 | 3.6 |
| | Gastrointestinal disease (%) | 4.4 | 4.0 | 0.0 |
| | Infectious disease (%) | 1.3 | 0.6 | 0.0 |
| | Metabolic disease (%) | 10.5 | 7.5 | 0.0 |
| | Cancer (%) | 3.7 | 1.8 | 0.0 |
| | Covid (%) | 3.3 | 3.2 | 0.0 |
| | Epilepsy (%) | 0.8 | 1.2 | 0.0 |
| | Pain (%) | 7.4 | 8.6 | 0.0 |
| | Other comorbidity (%) | 38.3 | 36.8 | 17.9 |
| Treatments | Antidepressant (%) | 23.0 | 23.1 | 0.0 |
| | Antihistamine (%) | 10.2 | 9.5 | 0.0 |
| | Antipsychotic (%) | 3.3 | 4.1 | 0.0 |
| | Benzodiazepine (%) | 25.0 | 24.9 | 0.0 |
| | Melatonin (%) | 6.7 | 7.1 | 0.0 |
| | Z-drugs (%) | 14.6 | 11.8 | 0.0 |
| | Opioid (%) | 0.6 | 0.6 | 0.0 |
| | RLS medication (%) | 1.2 | 0.6 | 0.0 |
| | Other medication (%) | 41.1 | 39.6 | 3.6 |

chronic insomnia and good sleepers. PSG recordings from patients were obtained as part of their clinical care whereas PSG recordings from good sleepers were obtained as part of their participation in research protocols (see details below). In both cases, participants provided written consent and approved the re-use of their data for research purposes. Chronic insomnia was diagnosed following the International Classification of Sleep Disorders, Third Edition (ICSD-3)[16], and defined as a complaint of persistent difficulties in sleep initiation, maintenance, or early morning awakenings for a minimum of three nights per week over a span of at least three months[15]. Patients were screened for obstructive sleep apnea (OSA) or periodic limb movement disorder (PLMD) by analysing PSG recordings and following existing guidelines (ICSD-3)[16]. All individuals with OSA and PLS were excluded from our analyses.

**Insomnia without SOSD (SOSD − )**. The insomnia without SOSD cohort comprised 624 patients (mean age 46.8 ± 14.1 years and 435 females (~70%)), who were retrospectively identified from the Sleep and Vigilance Center at Hôtel-Dieu Hospital in Paris, France. As part of routine clinical evaluation, these patients underwent PSG (either laboratory-based or ambulatory) to assess their sleep. We leveraged PSG recording to identify patients with significant objective disruption of their sleep as measured by a single night of PSG: sleep onset insomnia (SOL > 30 min), OR sleep maintenance insomnia (WASO > 30 min), OR early morning awakening insomnia (natural awakening > 1 h before desired wake time)[15,16]. Subjects meeting at least one of these criteria were categorized as "SOSD−."

**Insomnia with SOSD (SOSD+)**. 199 patients (mean age 40.8 ± 12.1 years and 148 females (~75%)) constituted the insomnia with SOSD group, identified through the same retrospective process at Hôtel-Dieu's

Sleep and Vigilance Center. SOSD denotes a disjunction between perceived and objective sleep patterns. This discrepancy can be established in various ways, for example by comparing PSG metrics and subjective reports on the same night. Here, we compared a general complaint of poor sleep (falling within the diagnostic criteria of Chronic Insomnia) and the PSG results obtained in one night of sleep recording. Patients were included in the SOSD+ group if they exhibited relatively normal PSG metrics: SOL < 30 min, AND WASO < 30 min, AND natural awakening < 1 h before desired wake time. Subjects meeting all these criteria were categorized as "SOSD+ " so that all insomnia patients were either SOSD− or SOSD+ . This approach does not take into account the variability in PSG outcomes across nights but is well adapted to clinical settings where, typically, only one night of PSG is performed.

**Control Group (GS)**. 104 participants (mean age 36.2 ± 11.9 and 32 females (~31%)) comprised the control group, recruited from prior VIFASOM studies[71–80], characterized by absence of subjective insomnia complaints, corroborated by PSG findings. The initial baseline night from prior protocols was consistently selected for analysis, conducted by the VIFASOM team. Control subjects underwent rigorous screening to exclude confounding factors such as sex, age, and comorbidities. Exclusions included individuals with OSA, PLMD, or TST outside the range of 4–8 h. In addition, subjects with chronic unbalanced pathologies, psychotropic treatments or those that could affect sleep were excluded from the protocols. Shift and night workers, people exposed to jet-lag or irregular sleep schedules (based on a 2-week sleep diary) were also excluded.

**Clinical characteristics**. In addition to undergoing PSG assessments, both SOSD− and SOSD+ subjects participated in comprehensive clinical

evaluations, which included a thorough medical history examination conducted by a qualified physician. Similarly, control subjects underwent a similar vetting process to ensure their suitability for study inclusion.

Comprehensive clinical data were systematically collected for all participants, encompassing various demographic variables such as sex and age, as well as anthropometric measures including height, weight, and body mass index. Additionally, detailed information regarding the participants' insomnia diagnosis, encompassing sleep initiation, maintenance, early morning awakening, and sleep state misperception, was recorded. The duration and nature of complaints, smoking status, current medication usage, and comorbidities were also documented (see Table 1 and Supplementary Table 1). Medications were categorized based on their known impact on sleep, while comorbidities were classified according to established associations with insomnia[15]. Disorders with fewer than 5 instances were grouped under the "other" category.

### PSG recordings and sleep scoring

The study employed multiple PSG devices, including NOX-A1 (Nox Medical, $n = 820$), ACTIWAVE (CamNTech Ltd, $n = 83$), MORPHEUS (Micromed S.p.A., $n = 13$), and SOMNOLOGICA (Medcare, $n = 11$). The NOX-A1, MORPHEUS, and SOMNOLOGICA systems included at least two EEG electrodes from C3, C4, F3, F4, O1, and O2, all referenced to their contralateral mastoid. The ACTIWAVE system used at least two of the EEG electrodes O1, O2, C3, F3, and FP1, also referenced to their contralateral mastoid. The number of EEG, EOG, and EMG derivations varied across devices, while respiratory parameters were recorded using NOX devices exclusively. All devices utilized in the study were validated and routinely employed in the sleep center and/or IRBA laboratory settings.

Following the completion of the night recording, PSG data underwent visual examination and scoring in line with the AASM guidelines[81] to generate individual hypnograms. The identification of arousals, leg movements, and respiratory events was also conducted. Arousals were characterized by EEG acceleration within epochs lasting between 3 and 15 s. This visual scoring process was executed by a sleep medicine specialist. Specifically, 30-second epochs containing EEG (mastoid-referenced), EMG, and EOG data were categorized into wakefulness (W), NREM stages 1 to 3 (N1, N2, N3), and REM sleep. As part of this retrospective analysis, all PSG files were subsequently re-scored by the same experienced sleep technician, ensuring consistency in sleep scoring across the dataset. From this secondary scoring, key parameters reflecting sleep macro-structure were extracted from individual hypnograms, including TST, WASO, SOL and duration of each sleep stage. For a detailed description of the sleep macro-structure see Supplementary Tables 5, 6, 7 and 8.

### Ethics

The research adhered to the ethical guidelines outlined in the Declaration of Helsinki of 1975, revised in 2001. Approval for the retrospective analysis of PSG data from both patients and controls was obtained from the local ethics committee (Comité de Protection des personnes (CPP) Ouest IV, Nantes, France). Patient data was pseudonymized and analyzed in compliance with legal regulations set forth by the Commission Informatique et Liberté (CNIL) in France. All ethical regulations relevant to human research participants were followed.

### Data analysis

#### Preprocessing.
EEG electrodes were referenced to its contralateral mastoid and then data was bandpass filtered in the 0.5–40 Hz range with a zero-phase finite impulse response with a Hamming window. Then, data was epoched in 30 s non-overlapping windows such that it matched the epochs used for the hypnogram. Only epochs with a valid sleep stage (W, N1, N2, N3, REM) were considered in the following analysis.

#### Spectral analysis.
The power spectral density (PSD) was computed for each EEG electrode using the multitaper method implemented in the *mne* Python library[82]. Given that different EEG devices had different sampling

rates, all the PSDs were projected into a common frequency range between 0.5 and 40 Hz with intervals of 0.5 Hz. PSDs were computed only for epochs that were scored as W, N1, N2, N3 or REM. Noisy epochs were usually not included because they were not scored as one of these 5 states if the signal quality did not allow for accurate scoring. We did not perform further data cleaning. Then, for each valid epoch, we extracted the normalized the power in 6 canonical bands (delta: 0.5–4 Hz, theta: 4–8 Hz, alpha: 8–12 Hz, sigma: 12–16 Hz, beta: 16–30 Hz and gamma: 30–40 Hz). Normalization was performed by dividing the PSD values by the sum of the whole PSD.

#### Data harmonization by hypnodensity estimation.
The data used in this work was recorded with devices that differed in the number of electrodes, the electrode layout and the sampling rate. This diversity hinders the possibility of directly analyzing all the subjects in a common space. To overcome this limitation, and to use a sample as big as possible, we developed an harmonization procedure based on estimating hypnodensities (HD)[38]. A HD is a probability distribution supported on the 5 different stages and quantifies the probability of each sleep stage to be present in a given epoch. This approach conveys more information than traditional hypnograms and provides a continuous representation of sleep dynamics.

To estimate the HD we developed an analytical pipeline based on massive feature extraction and machine learning. We used the *catch22* Python library[42] to characterize each available EEG and EOG channel with 22 features (see the original publication for a detailed description). These features have been tailored to efficiently characterize a wide diversity time-series by capturing complementary and non-redundant aspects of the signals. Then, for example, if a recording had 3 EEG + 1 EOG channels, we would obtain $22 \times 4 = 88$ features. This analysis was performed for each epoch and the number of features changed for each subject depending on the number of EEG and EOG channels.

Next, we used an intra-subject leave-one-out classification approach to predict the sleep stage of each epoch using these *catch22* features and the respective hypnograms as input for a XGBoost multiclass classification algorithm[83] (see below for algorithm parameter details). For example, if a subject had 1000 epochs, we used the features and respective sleep stages of 999 epochs to train the classifier and tested on the remaining epoch. This procedure was repeated for each epoch. Finally, to obtain the HD, the classifier's prediction scores were transformed into class probabilities by the Platt scaling calibration procedure[84]. This way we exploited and integrated the information present both in the PSG and in the hypnograms (scored by experts), enabling for the projection of every epoch of each subject into the same 5-dimensional space.

#### Intrusions and instability.
As HD are probability distributions, tools from information theory can be leveraged to characterize them in a principled and interpretable way. Here we wanted to characterize two aspects of sleep dynamics: i) intrusions of other stages and ii) instability. We quantified them using two well known measures, the entropy (H) and the Kullback-Leibler divergence ($D_{KL}$)[85], respectively. Entropy quantifies the uncertainty or surprise associated with the measurement of a given variable, while the $D_{KL}$ measures the difference between two probability distributions. When base 2 logarithms are used, their units are maybits. In detail, for a random variable $x$ with discrete probability distribution $P(x)$, the entropy $H(x)$ is defined as:

$$H(x) = -\sum_{i \in \Omega} p_i \, log_2(p_i),\tag{1}$$

where $x$ corresponds to the HD of a given epoch, $\Omega = \{$W, N1, N2, N3, REM$\}$, i.e., the 5 possible sleep stages, and $p_i$ is the probability of each possible stage. For discrete probability distributions, entropy is a non-negative measure that is maximized when all the stages have the same probability (uniform distribution) and is 0 when a given stage has the maximum probability (a Dirac distribution). Then, an HD with large

intrusions of other stages will have high entropy (as the HD approaches a uniform distribution), while an epoch with small intrusions will have small entropy (as the HD approaches a Dirac distribution). This way, entropy is directly proportional to the level of intrusions and inversely proportional to the 'purity' of a given HD. In other words, the more certain we are about a specific stage, the lower the entropy.

Then, to measure the instability of an epoch with respect to the next one within the same sleep stage, we used the $D_{KL}$ to quantify the difference between the two consecutive HD. In detail, for two probability distributions $P(x)$ and $Q(x)$ (i.e., the HD of two consecutive epochs) the $D_{KL}$ follows:

$$D_{KL}(P||Q) = \sum_{i \in \Omega} p_i \, log_2 \left( \frac{p_i}{q_i} \right), \qquad (2)$$

where $p_i$ and $q_i$ is the probability of the stage $i$ for the $P$ and $Q$ HD, respectively. If the HD of two consecutive epochs are equal, i.e., maximal stability, the $D_{KL}$ is zero, while it increases as the two HD become different. Then, the $D_{KL}$ is directly proportional to the instability of two consecutive epochs. In other words, if a big transition of the dynamics occurs from one epoch to the next one, the $D_{KL}$ will be big.

Finally, for each subject, the average and standard deviation of both metrics were computed across all the epochs associated with a specific stage. This yielded a set of 16 features (the entropy and the $D_{KL}$ for 4 stages and their respective average and standard deviation; N1 was not considered as a feature). These features were subsequently used as inputs to train a classifier for discriminating between GS, SOSD− and SOSD+, and also for a regression to predict TST, WASO, SOL and SE.

**Feature selection**. To enhance interpretability and diminish noise, we conducted a feature selection process on the set of information theoretic features. For both classification and regression tasks, we employed the minimum redundancy maximal relevance (MRMR) algorithm[86] to rank the features. This algorithm assesses feature relevance using the performance of a random forest, while redundancy is measured by the mean correlation with the rest of the features. Subsequently, the features were progressively included as inputs for cross-validated classification or regression based on their MRMR scores. To determine the final set of features, we computed the cross-validated balanced accuracy (for classification) or root mean squared error (RMSE, for regression) for each feature subset. We then identified the optimal subset as the point where further inclusion failed to significantly enhance performance (measured via a Wilcoxon rank sums test).

**Machine learning classifier and regression**. All the classification and regression tasks were performed using a XGBoost algorithm[83]. XGBoost is a type of boosted ensemble of decision trees that have been proven accurate and efficient for many real-world applications. For multiclass problems (stage prediction and three-class prediction of diagnose) we used the following parameters: a *softprob* objective function, a *logloss* evaluation metric, a learning rate of 0.1, maximum tree depth of 6, a subsampling of 80% of subject, a feature subsampling of 80% and a regularization gamma parameter of 0.1. For the case of binary classifications (SOSD− vs SOSD+), the parameters were the same, with the only difference that the maximum tree depth was 3. For the regression tasks the objective function was the MSE and the rest of the parameters were the same as for classifications. These parameters were selected to avoid overfitting and, as they yielded good results, no hyperparameter optimization was done. For classification and regression, performance metrics were obtained from a repeated (60 iterations, unless other is specified) 5-fold stratified cross-validation procedure by generating 5 random partitions with approximately the same number of samples (subjects for diagnose prediction, epochs for sleep stage prediction). Then, the model was fit using 4 of these partitions, and its performance (balanced accuracy and confusion matrix for classification, and RMSE for

regression) was tested using the group that remained, and repeated this process until all partitions were used for testing and training. For the stage prediction (see *Data harmonization by hypnodensity estimation*), a leave-one-out procedure was used, so balanced accuracies and confusion matrices were computed after predicting each epoch separately.

**Matching samples by size, sex, age and body-mass index**. To verify that imbalances in size, sex, age and body-mass index (BMI) could be biasing the results (see Supplementary Table 1), we performed a sub-sample analysis where these variables were controlled. In the case of size, we used the smallest sample size among classes (GS = 104 for the three-class taks and SOSD+ = 199 for the SOSD+ vs SOSD− task) and took 30 equal sized random subsamples of the other class(es). This way the algorithm was trained with a perfectly balanced sample in terms of size.

For the binary classification between SOSD− and SOSD+, we matched the samples in terms of sex, age and BMI. To do so, we randomly selected a subsample with perfect balance in terms of sex and size (38 females, so 76 subjects for each condition) and used a Kolmogorov-Smirnov test to evaluate whether SOSD− and SOSD+ significantly differed both in age and BMI. If both tests were rejected (*p*-value > 0.1), the subsample was considered as balanced in size, sex, age and BMI. We generated 500 of these balanced subsamples and used them for the binary classification task.

**Interpretation of information theoretic measures**. To interpret the entropy and $D_{KL}$ in terms of traditional sleep features (PSD and stage transitions) we performed correlations between entropy and the EEG spectral power and between the $D_{KL}$ and self-transitions of sleep stages, respectively. For entropy and $D_{KL}$, we took the power in the 6 canonical bands (delta: 0.5–4 Hz, theta: 4–8 Hz, alpha: 8–12 Hz, sigma: 12–16 Hz, beta: 16–30 Hz and gamma: 30–40 Hz) of all the epochs related to a specific stage and correlated to the entropy and $D_{KL}$ values associated with those epochs. We obtained one correlation value for each stage and subject. For the $D_{KL}$, we computed the transition matrix for each subject and extracted the self-transition associated with each stage and correlated with the average $D_{KL}$ of that stage, obtaining one correlation value for the whole sample and for each condition.

**Statistics and reproducibility**

A Wilcoxon rank sums non-parametric test was used both to evaluate the differences between the information theoretic values and to compare the distributions of performance values obtained from the classifiers. Pearson correlation was used to compute correlation between entropy and $D_{KL}$ and the macro and microstructural features of PSG. All p-values were corrected for multiple comparisons with the Bonferroni method. Group sample sizes correspond to SOSD−: $n = 624$, SOSD+: $n = 199$ and GS: $n = 104$; Each sample corresponds to a different participant. No replicates were used.

**Reporting summary**

Further information on research design is available in the Nature Portfolio Reporting Summary linked to this article.

## Data availability

Due to protection of personal privacy, the clinical database used in this work cannot be publicly available. However, the data can be provided by DL pending scientific review and a completed material transfer agreement. Requests for the PSG recordings and associate metadata should be submitted to damien.leger@aphp.fr.

## Code availability

Computational codes to extract hypnodensities and compute intrusions and instability from an example dataset are available in a Zeonodo repository[87].

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

## Acknowledgements

This survey has been supported by an unrestricted grant of the AID (Agence Innovation Défense) to the RAPID PANDORE-IA project and received financial support from Agence Nationale de la Recherche (ANR-22-CE37–0006-01) and the European Research Council (ERC-StG SleepingAwake 101116748).

## Author contributions

Conceptualization: R.H., D.L., T.A.; Methodology: R.H., F.C., A.A., U.H., M.C., D.L., T.A.; Investigation: R.H., F.C., A.A., T.A.; Visualization: R.H.;

Supervision: D.L., T.A.; Writing-original draft: R.H., F.C., A.A., U.H., D.L., T.A.; Writing-review & editing: R.H., F.C., A.A., U.H., D.L., M.C., T.A.

## Competing interests

The authors declare no competing interests.
