## [Transparent Peer Review file · Communications Biology]

A continuous approach to explain insomnia and subjective-objective sleep discrepancy

Corresponding Author: Dr Thomas Andrillon

Version 0:

Reviewer comments:

Reviewer #1

(Remarks to the Author)

In this manuscript on a large retrospective database of patients with insomnia and healthy controls (HC), the authors use hypnodensities (HD) provided by a machine-learning algorithm to obtain more information on group differences, also including subgroups of insomnia characterized as sleep stage misperceivers (SSM) and objective insomniacs (INS). HD are probabilities (p) for each of the 5 wake and sleep stages.

The database originally consists of 927 single-night polysomnographies (PSG) and was used for previous publications (eg Andrillon et al. 2020).

The authors find HD group differences mainly in expert-staged wake epochs, namely that SSM shows more "sleep intrusions", meaning that the ML algorithm was less sure about the stage to assign, operationalized by the authors as a high "entropy" across the 5 p -values for all stages of an epoch staged as wake.

A second information theoretical measure used by the authors is the difference in HD between successive epochs of a given expert-staged sleep stage, operationalized by the Kullback-Leibler divergence (DKL). The means and standard deviations of these two values for each epoch per stage were then used as feature vectors for the classification of group and also to predict classical sleep parameters such as total sleep and wake times. For example, larger sleep time and, correspondingly, shorter wake time was found related to increased mean entropy in wake.

First of all, this is an extensive work on one of the largest collections of subjects with insomnia that is available. The work is also addressing a very relevant clinical question and a persistent enigma of sleep research - the discrepancy between PSG sleep quality and the subjective perception of sleep.

There are, however, serious problems with this work that indicate that the authors lost track of the interpretation of their data early on in this tedious endeavor.

In particular, what the authors find is circular: Obviously they have found a correlate of having few wake epochs within a given night. And then they see differences between SSM and INS, which are subgroups selected on exactly that: SSM, by definition, have few wake epochs, and INS have many.

The more technical basic mistake is to train the algorithm for stage classification separately on each single subject. This is synonymous with a large chance of bias for this specific subject's EEG rather than defined universal characteristics used for staging, and more importantly, this single night's composition of sleep stages defines the weight each stage has in the optimization - nights with few wake epochs will have systematically worse wake staging. Not because of any specific signal property but just because wake is not so important if there are fewer wake epochs. This can easily explain the higher "entropy" in the HD when staging wake epochs in PSGs with few wake epochs.

Sadly, the data must be very carefully examined for this effect and the interpretation adapted correspondingly. Most probably, the authors have not found any specific property of insomnia at all.

Further general comments:

Structure: The authors tried to separate methods from introduction and results but only achieved that partial information not helpful to understand what was really done appears in both introduction and results. The number of subjects examined, for example, is given in all three sections with varying numbers, but still it is unclear how many datasets were really used after exclusion of those in which the ML staging didn't work - which is the important number for the conclusions. Similarly, results appear in the introduction ("Finally, our approach was also sensitive to insomnia severity ...")

Definition of insomnia: Insomnia is defined by subjective complaint in all of the relevant manuals. Even the subclass "sleep state misperception" was removed in DSM-5 because it is not reliable. Therefore, talking about "insomnia severity" in relation to PSG is simply wrong. Look at the Insomnia Severity Index (ISI) to see how that is defined.

The same mistake is seen in multiple places. A big one is, in the methods section, the definition of the SSM group: "Subjects failing to meet any ICSD-3 criteria for insomnia were classified as "SSM."" So then they were not an insomnia subgroup?

Be more specific when mentioning sleep stages: It should be clearer that expert staging is mostly meant in wordings such as "during wake". Consider that there are sleep stages as classified by the ML algorithm as well.

Further specific comments:

"Increased intrusions in insomnia": "For this reason, N1 was excluded from the following analysis." And still, N1 is described to be included in all feature vectors (one of 5 or 6 sleep stages). Was it excluded or not?

Discussion:

"our algorithm is fairly simple and can be opened up": Instead of "open up", a commonly used term nowadays is "explainable", as in "explainable AI/ML".

Data analysis:

"Intrusions and stability": The authors in several places describe their DKL measure as "stability" but it is actually *instability*, as it measures the mean differences between successive epochs. For example also on line 283: "The average intrusions and stability in wakefulness had the largest correlation with WASO and SE, with a negative correlation for the former and a positive for the latter" which would not be understandable because average intrusions and stability should be anticorrelated - but what the authors call "stability" is DKL, ie instability.

It is unclear, BTW, how "successive epochs" are defined for DKL. I assume that this is after filtering for sleep stage, meaning that both differences between actually successive epochs and differences between epochs of the same sleep stage separated by other stages are included. The latter pairs should be excluded.

Ethics: Wouldn't the GDPR be more relevant? Legally, local guidelines cannot deviate from it.

Reviewer #2

(Remarks to the Author)

This paper uses machine learning to study hypodensities and entropy in insomnia individuals (INS). Its main aim is to quantify/qualify sleep state misperception (SSM). By comparing individuals (INS) presenting sleep difficulties to individuals presenting SSM and good sleepers, authors have set forward new investigation methods which contribute to our understanding of SSM. In fact, throughout their paper, results showed that SSM have sleep intrusion during wake in the night while INS have wake intrusions during sleep. Furthermore, it appears that these results indicate that while SSM individuals do not differ from GS in many respects, they do when the microstructure of sleep is considered. Authors conclude, with adequate emphasis that sleep, and its mismeasurement might be distributed along a continuum and that it is only by deciphering the EEG that one can justly estimate the contribution of stability and intrusions to sleep quality.

This paper is one of the few I have been truly excited to read and review during my career. As has been observed in my own lab, it is by looking at the EEG and using fine tune methods that one can really quantify sleep/wake disruptions in INS. Insomnia disorder pertains to a large array of symptoms and encompasses the mere subjective complaints in addition to showing a wide range of heterogeneity within suffering individuals. This paper is very enlightening and set the way forward

for using similar analyses with other groups of disorders and for, by example, attending to the contribution of one disease/disorder or another. It also suggests that we should stop trying to classify individuals into categories which I find reassuring. Those seemingly on the 'verge' of being into one or the following categories is subjected to seemingly arbitrary cut-offs, and it should not be. Thanks for this brilliant work.

Here are few comments which should be addressed, if not, seriously considered.

- 1) It appears that not much allusion is made to the fact that INS often show good and bad nights (the disorder does specify 3 or more nights per week). By defining sleep quality as they did, do authors feels that their results would differ according to bad or good nights? If so, how could we explain those?
- 2) Authors also suggest that results could be geared towards bettering treatment. They offer very poor alternatives or suggestions in that regards. They should be more explicit about what they suggest.
- 3) Any ideas on the neuronal mechanisms underlying sleep/wake mismeasurements besides the theory of local cortical vigilance states?
- 4) Sleep quality is a difficult concept to grasp, define. For one it will mainly be the 'type' of sleep difficulties while for the other, the number of arousals during the night. What would authors suggest being different between those different types?
- 5) The reviewer's understanding is that the selected EEG was 'very clean'. What does it mean? Does it mean that stage transitions with movements or short awakenings were discarded? It is understood that AASM criteria were used for the scoring. How many epochs were 'un-scorable' and how many were rejected?
- 6) It appears that SSM were twice as subjected to cardiovascular diseases than INS. How could this influence the data?

Reviewer #3

(Remarks to the Author)

Thank you for the opportunity to review this interesting paper approaching the phenomenon of SSM in Insomnia Disorder using two measures of intrusions and stability of sleep through advanced machine learning techniques. The rationale and the aim of the study are paramount in the study of sleep perception in insomnia patients, and the methodologies applied could provide new and important information on the electrophysiological processes underlying SSM. However, some methodological choices severely limit the quality of the paper.

The main methodological limitation of the study is the definition of the SSM groups as subjects who complained of insomnia symptoms during the clinical evaluation but did not exhibit PSG abnormalities during the recording night. However, this definition does not account for the first-night effect and, especially, for the night-to-night variability in insomnia symptoms (Buisse et al., 2010; Newell et al., 2012). Moreover, how long has it been since the patient complained of symptoms of insomnia during the clinical visit and when PSG was performed? The degree and definition of the SSM are very difficult to establish, and the standard way to measure the SSM degree is to collect the subjective sleep times perceived by the individual the morning following the PSG recording, a few minutes after the final awakening to ensure an accurate and unbiased memory of the perception of night sleep (Castelnovo et al., 2019). In my opinion, the way used by the authors to discriminate insomnia patients with and without SSM is not appropriate from a methodological standpoint. Another point regarding the insomnia disorder group, is the definition of inclusion/exclusion criteria. Has the exclusion of other sleep disorders been established? How? explain in the text. Moreover, as a minor point, in the Objective Insomnia paragraph in the methods section authors reported that delineated three subtypes of insomnia based on the objective PSG indices of SOL (i.e., Sleep onset insomnia subtype), WASO (i.e., Sleep maintenance insomnia subtype), and the morning awakening (i.e., early morning awakening insomnia subtype). However, the analyses do not account for this subtyping. What was the purpose of this distinction?

As a second methodological limitation, the study is based on the analysis of sleep intrusion during wake epochs and, vice versa, of wake intrusion during sleep epochs. However, the methodological approach used by the authors does not allow to distinguish which type of sleep intrusions are present during wake, and in which specific sleep stages the waking intrusion occurs. In other words, from a physiological and processual point of view, it is totally different if a wake intrusion occurs during an epoch of N1 or N3 sleep, and, on the other hand, it will be very different if, during a wake epoch, there will be an intrusion of REM sleep or SWS. Referring generally to wake or sleep intrusions without a characterization of these physiological states is, in my opinion, not informative in the process of SSM and is sometimes misleading. For example, you defined levels of entropy as "high HD entropy indicates a high level of intrusion (proximity to a uniform distribution), high entropy during wakefulness implies sleep intrusions, while high entropy during sleep implies wakefulness or intrusions from other sleep stages". However, when you reported findings in the three groups, results of larger entropy during N2 and N3 are interpreted to "an increase in wake probabilities". But are we sure that these intrusions are of wake and not of other sleep stages if the analysis is aspecific to the type of intrusions that occur? The same lack of characterization is detrimental in interpreting the results of sleep stability. Indeed, it is very different if a period of N3 is followed by epochs of N2, REM sleep, or wake. Without this type of distinction, these results are open to speculation not supported by the findings. In these terms, another point is the interpretation of results about stability. Authors report: "... only SSM shows an increased instability in wakefulness compared to GS. Again, these results are consistent with the condition of SSM as a disruption of intra-sleep wakefulness rather than sleep itself.". My question is: is it permissible to interpret the result of wake periods interrupted by sleep as disruption of intra-sleep wakefulness? In my opinion, this result may be interpreted as a wake-sleep pattern fragmented in SSM patients, in terms of difficulty in reaching stable sleep. Moreover, it could be interesting and informative to report at what time of night intrusions and stage shifting occur (i.e., it is very different if an intrusion of N3 sleep during wake occurs at the beginning or at the end of the night, or if the wake instability occurs at the sleep onset or in the middle of the night). Analyses of sleep cycles may be informative.

Other limitation points regard the methods used to record sleep and extract PSG indices.

- As a retrospective study, the number of EEG derivations varied across devices used to record sleep. Please report, also in supplementary materials, the different EEG montages used in the study.

- In the Preprocessing paragraph is not reported if an artifact rejection was carried out for the power spectral analyses. Please, specify whether and how (i.e., manual or automatic) it was carried out.
- Regarding TST and duration of each sleep stage, absolute (i.e., in minutes) or percentage indices were used?
- In the Spectral analysis paragraph, it is not reported if the power is relative or absolute and if the spectral analyses were carried out for single bands, and for single sleep stages or NREM/REM sleep. Please, specify these issues in the text.
- The text does not present a description of the sleep macrostructure in the three groups examined. Please report PSG macrostructure indices and the statistical between-group differences, also through a table.

Besides the mentioned methodological limitations, the paper, in my opinion, presents other conceptual, interpretation, and presentation problems.

First of all, the way in which results of “Increased intrusions in insomnia: sleep intrusion in SSM and wake intrusion in INS” are reported makes them difficult to understand. Especially the first part of the description of the results is not accompanied by a clear image to help to understand the findings. I guess that the results described refer to Figure 2 A; however, this is never cited in the text. Please, provide a better explanation of these findings and pay attention to the figure.

In the “Sleep intrusions and wake stability predict diagnosis and sleep quality” paragraph authors report: “The average intrusions and stability in wakefulness had the largest correlation with WASO and SE (Figure 4C), with a negative correlation for the former and a positive for the latter. Accordingly, our results suggest that the level of intrusions and stability in wakefulness can not only discriminate between sleep groups but also predict the severity of insomnia.” In my opinion, this interpretation is tautological. Indeed, the negative correlation between the average intrusion and stability in wakefulness and the amount of WASO is an effect of the wake stability, on the other hand the positive correlation with the SE is the other side of the coin due to the increase of sleep. Moreover, why the authors interpreted these results as predictors of the severity of insomnia?

It would be useful to include correlation coefficients when reporting correlational results in the text.

The way in which Figures 4C, 5 A, B, C, D, and F are shown does not allow accurate reading of the results. About Figures 4C, what are the significant correlations (report p plus rho)? Why do authors sometimes use the two decimal places after the comma and in other cases 3, 4 or 5? It would be necessary to be uniform. Considering Figures 5A, B, C, D, and F, why did the authors graph correlational results through bar plots and not scatter plots? Please provide scatter plots of correlational results, so that the distribution of the single points of the correlation can be observed.

Minor revisions

- Introduction session:

o When you report the reference number 16 there is a double bracket.

o You have rightly reported that “SSM is present in all sleep disorders” without mentioning that, although to a lesser degree, this is a phenomenon also recognized in good sleepers. I recommend a brief mention.

o In the period “Based on this relationship, the current consensus is that the detection of certain...” you defined the EEG acronyms. However, had already defined a few lines above.

o When you addressed the topic of the current lack of characterization and knowledge of the mechanisms underlying SSM and sleep perception due to the limitations of PSG analysis, I suggest citing a recent work where a topographical analysis was carried out: “Fasiello, E., Gorgoni, M., Galbiati, A., Sforza, M., Berra, F., Scarpelli, S., ... & De Gennaro, L. (2024).

Decreased Delta/Beta ratio index as the sleep state-independent electrophysiological signature of sleep state misperception in Insomnia disorder: A focus on the sleep onset and the whole night. *NeuroImage*, 298, 120782.”

- Results session

o NREM and REM acronyms are never defined in full in the text. Please, define.

o Figure 2 is titled “Intrusions and stability on insomnia with and without SSM.” However, the figure illustrates also findings in GS. Please, specify in the figure description.

o The Figure 2A is not cited in the text.

o In the “Sleep intrusions and wake stability predict diagnosis and sleep quality” paragraph the acronyms of TST, SOL, WASO, and SE are defined. However, these acronyms were yet defined in the text. Please uniform.

- Discussion session

o Uniform the sleep state misperception term in the first period of the paragraph with the acronym SSM as in the rest part of the text.

o Authors defined the PSG acronym in the first period of the paragraph, but it is yet defined in the text. Please, uniform.

- Materials and methods session

o In the Participants paragraph authors defined the PSG acronym, but it is yet defined in the text. Please, uniform.

o Table 1: provide indices of statistical differences between the three groups, in addition to descriptive results

o The description of the recruitment procedure as a single paragraph does not help the reading. Please report the specific recruitment procedures for the single groups in the relative paragraphs (i.e., Objective Insomnia, Sleep Sate Misperception, and Control Group).

o Please, cite in the text original studies in which the good sleepers were recruited (i.e., VIFASOM).

o In the Control Group paragraph recruited individuals were defined as “patients”. Please correct this. Moreover, the authors report that GS underwent rigorous screening to exclude confounding factors such as comorbidities. Which types of comorbidities were excluded?

Version 1:

Reviewer comments:

Reviewer #1

(Remarks to the Author)

The authors have responded well to the extensive review, clearly enhancing the manuscript. I have no further suggestions.

Reviewer #2

(Remarks to the Author)

In my opinion, the authors have adequately address all reviewers comments. Congratulations on a great addition to the field of insomnia.

Reviewer #3

(Remarks to the Author)

Thank you to the authors for responding promptly to all my comments. However, sadly, all the reviews made by the authors in accordance with the requests of the reviewers do not fill the enormous methodological gap of this work. Specifically, regarding question number 3, "How long has it been since the patient complained of symptoms of insomnia during the clinical visit and when PSG was performed?" authors have reported that between the subjective report of sleep disturbance and the objective sleep recording have passed an average of 6 weeks. This approach is inadmissible to measure the subjective-objective sleep discrepancy; for example, it does not consider that in perspective of the night-to-night variability, how many of the tested patients would have reported sleeping well at night from PSG recording. All work is presented with a focus on a phenomenon (i.e., SSM, or the subjective-objective sleep discrepancy) that has been evaluated and conceptualized in an inaccurate way.

General comment:

Reviewer #2 and #3 pointed out that the definition of our insomnia sub-types (INS and SSM) differs from some previous studies. Indeed, we define Sleep State Misperception by relying on the subjective complaint of the patients to assess the presence of insomnia (as advised by international guidelines) and a single night of PSG recordings (as sometimes done in sleep clinics) to determine if this complaint is in line with significant changes in objective metrics of sleep. However, we did not test (for lack of the relevant data) whether there was a misperception during the night of PSG itself, i.e. a discrepancy between the PSG recordings and the subjective report of individuals on the same night. Yet, from a clinical point of view, our approach is closer to the clinical practice since patients consult because of a complaint and then undergo a single night of PSG.

However, to clarify this point, we modified our article accordingly. We also modified our clinical labels to avoid confusion with previous studies. We use the following group labels : (1) Good Sleepers (GS), (2) SODS+ (insomnia patients with subjective-objective sleep discrepancy, formerly SSM), (3) SODS- (insomnia patients without subjective-objective sleep discrepancy, formerly INS). This change has the advantage of distinguishing between the phenomenon of SODS and the individuals to whom we attributed an SODS label (SODS+ group), a distinction our initial terminology did not make. It also stresses that both groups were indeed diagnosed with insomnia, a fact that the distinction between INS and SSM was perhaps obscuring as noted by Reviewer #1. Finally, we preferred the term SODS to SSM, based on the current state of the art outlined in a recent review (Stephan & Siclari, 2023). Besides, SSM suggests that the discrepancy between subjective and objective reports stems from a misperception (so an erroneous subjective assessment, which fits less with our current work since participants were not asked to provide their perception of their own sleep on the night of the PSG recordings.

Overall, we believe this change addresses comments from all Reviewers and will make our work clearer.

Reviewer #1 (Remarks to the Author):

1. In this manuscript on a large retrospective database of patients with insomnia and healthy controls (HC), the authors use hypnodensities (HD) provided by a machine-learning algorithm to obtain more information on group differences, also including subgroups of insomnia characterized as sleep stage misperceivers (SSM) and objective insomniacs (INS). HD are probabilities (p) for each of the 5 wake and sleep stages. The database originally consists of 927 single-night polysomnographies (PSG) and was used for previous publications (eg Andrillon et al. 2020). The authors find HD group differences mainly in expert-staged wake epochs, namely that SSM shows more "sleep intrusions", meaning that the ML algorithm was less sure about the stage to assign, operationalized by the authors as a high "entropy" across the 5 p -values for all stages of an epoch staged as wake. A second information theoretical measure used by the authors is the difference in HD between successive epochs of a given expert-staged sleep stage, operationalized by the Kullback-Leibler divergence (DKL). The means and standard deviations of these two values for each epoch per stage were then used as feature vectors for the classification of group and also to predict classical sleep parameters such as total sleep and wake times. For example, larger sleep time and, correspondingly, shorter wake time was found related to increased mean entropy in wake. First of all, this is an extensive work on one of the largest collections of subjects with insomnia that is available. The work is also addressing a very relevant clinical question and a persistent enigma of sleep research - the discrepancy between PSG sleep quality and the subjective perception of sleep.

We appreciate the reviewer's comments and his detailed summary of our work. We would like to first clarify that we did not use the same data as in our previous publication (Andrillon et al 2020). We more than doubled the initial database (927 vs. 436 recordings). The inclusion centre is however indeed the same (Hotel-Dieu Hospital).

2. There are, however, serious problems with this work that indicate that the authors lost track of the interpretation of their data early on in this tedious endeavor. In particular, what the authors find is circular: Obviously they have found a correlate of having few wake epochs within a given night. And then they see differences between SSM and INS, which are subgroups selected on exactly that: SSM, by definition, have few wake epochs, and INS have many.

We strongly disagree with this characterization of our work and we do not think that we “lost track of the interpretation of [our] data”. Yet, the Reviewer’s comment on the circularity of our analyses is very important to address here and in our revised manuscript precisely because our analyses are not circular and it is important to convey this message clearly. We believe that the Reviewer’s argument stems from confusion about the nature of our analyses, which we will seek to clarify here and by making modifications in the article.

In stating that our findings are circular and trivial, the Reviewer wrote: “Obviously they have found a correlate of having few wake epochs within a given night”. We report many changes and correlates of SOSD/SSM not just one so it is unclear what the Reviewer had in mind precisely. Besides, many of these correlates cannot be directly linked to the presence of few wake epochs within a given night. We therefore do not think that our results are circular as we will develop below.

First, we find differences between SOSD+ and GS individuals although they have similar sleep architecture and proportion of wake epochs (see Supp. Table 7). Namely, SOSD+ patients have higher entropy in wakefulness and N2 than controls and lower dKL in N3 and REM sleep. These results on entropy are thus not a mere reflection of the proportion of wake epochs.

Second, to fully address the Reviewer’s comment , we run a quantile analysis (for the median) using the normalized frequency of each sleep stage as a covariate to test if the group differences held when taking into account the frequency of each sleep stage. We still found significant group effects in wake for all pairwise comparisons. We show the results in the attached table, that now is part of the supplementary material (Supplementary Table 3). Consequently, the results on entropy and DKL can’t be solely explained by the differences in the number of wake epochs. We now cite this table after presenting the results of Figure 2 to emphasize that these results hold even after controlling for stage normalized frequency.

Entropy						
GS vs SOSD-		GS vs SOSD+		SOSD- vs SOSD+		
Coeff	P-value	Coeff	P-value	Coeff	P-value	

W	-0,07734	0	-0,13637	0	-0,05902	0
N2	-0,03243	0,13842	-0,01496	1	0,01747	1
N3	-0,05651	0	-0,03424	0,02231	0,02227	0,02902
REM	-0,02724	0,40865	-0,02137	1	0,00586	1

3. The more technical basic mistake is to train the algorithm for stage classification separately on each single subject. This is synonymous with a large chance of bias for this specific subject's EEG rather than defined universal characteristics used for staging, and more importantly, this single night's composition of sleep stages defines the weight each stage has in the optimization - nights with few wake epochs will have systematically worse wake staging. Not because of any specific signal property but just because wake is not so important if there are fewer wake epochs. This can easily explain the higher "entropy" in the HD when staging wake epochs in PSGs with few wake epochs.

We disagree with the Reviewer's description of this methodological choice as a "basic mistake". We opted for this approach precisely because it allows building personalized models and avoiding a pattern of results that could be directly linked to an algorithm not adapted to a specific clinical population. This is now highlighted in the Abstract and in the Introduction: "[...] we used a personalized model of sleep architecture. From this personalized model, we extracted hypnodensities to train a cross-subject algorithm for the detection of insomnia [...]" (p. 3).

Furthermore, we considered this caveat in our initial submission and replicated our classification results using a model trained on a separate dataset (U-Sleep) and tested on our data. As demonstrated in the Supplementary Figure 2, when using a common and 'universal' model (U-sleep), the classification performance drops but still remains above the chance level. This means that hypnodensities, entropy and dKL derived from a common model can still separate the three groups, answering the Reviewer's concern. We stressed this point in the revised manuscript (p. #):

"This additional analysis was also designed to test if an algorithm trained on a different dataset would extract hypnodensities that allow the classification of the three groups even when the model is not trained on the PSG recordings themselves. Following the same feature selection and inclusion procedure, the hypnodensity derived from this external model yielded lower but above chance classification performance (Supplementary Figure 2)."

Finally, even if the classification accuracy obtained with U-sleep was significant, our personalized approach performed much better (0.1 increase in AUC value), obtaining an unprecedented classification performance for the classification of insomnia and its subtypes (see the review of Stephan & Siclari 2023). Thus, the personalized information that we obtained with our model provides more discriminative information in a cross-subject model than a trained 'universal' approach, illustrating the relevance of our methodological approach.

4. Sadly, the data must be very carefully examined for this effect and the interpretation adapted correspondingly. Most probably, the authors have not found any specific property of insomnia at all.

Given our replies to comments 1 to 3, we must disagree with the reviewer. First, we show that the difference between S OSD- and S OSD+ individuals is still significant when factoring in the proportion of wake epochs. Second, we report differences between GS and S OSD+ which cannot

be attributed to trivial differences in sleep architecture since this architecture is similar. Third, we show that we can still classify the three groups based on hypnodensities and hypnodensity-derived metrics derived from a common generic model. In conclusion, our results are robust to the confounding factors raised by the reviewer and our work shows an unprecedented classification performance that has not been achieved by previous approaches using macrostructural features as the stage frequency (see also our previous work in Andrillon et al. Sleep Medicine 2020).

Further general comments:

5. Structure: The authors tried to separate methods from introduction and results but only achieved that partial information not helpful to understand what was really done appears in both introduction and results. The number of subjects examined, for example, is given in all three sections with varying numbers, but still it is unclear how many datasets were really used after exclusion of those in which the ML staging didn't work - which is the important number for the conclusions. Similarly, results appear in the introduction ("Finally, our approach was also sensitive to insomnia severity ...")

We thank the reviewer for the careful reading. We have now removed the sample sizes from the introduction and kept them only in the results and methods section for completeness. It is however incorrect to say that we provided different numbers across sections. We checked our initial submission and revised manuscript and the sample size for the different groups are always the same.

It is however true that we only presented data for the subjects where the algorithm had an accuracy higher than 0.4. This means that 115 recordings were discarded compared to the total available data. These subjects were excluded from all analyses and are not included in the Results, Figures and Tables. This is now better described in the Methods section (p. 17):
"The total number of available recordings was 1042. However, 115 recordings were discarded because of noisy data or because the hypnogram was predicted with a balanced accuracy lower than 0.4 (see below). These recordings were excluded from all analyses, leading to a total sample size of 927 recordings. These excluded files are not included in the reported sample sizes or results."

Finally, as the Reviewer mentions, we provide a summary of our findings at the end of the introduction. This is intentional and rather common in scientific articles but we are open to suggestions from the Editor in accordance with the Journal's formatting guidelines.

6. Definition of insomnia: Insomnia is defined by subjective complaint in all of the relevant manuals. Even the subclass "sleep state misperception" was removed in DSM-5 because it is not reliable. Therefore, talking about "insomnia severity" in relation to PSG is simply wrong. Look at the Insomnia Severity Index (ISI) to see how that is defined. The same mistake is seen in multiple places. A big one is, in the methods section, the definition of the SSM group: "Subjects failing to meet any ICSD-3 criteria for insomnia were classified as "SSM."" So then they were not an insomnia subgroup?

We agree with the reviewer that the definition of insomnia is based on subjective criteria only, and we explicitly refer to this and the ICSD-3 criteria in our initial submission ("*chronic insomnia is defined on subjective reports alone*", p. #). However, we agree that we chose confusing terms to characterize how we incorporated results from PSG recordings. We have revised the Methods section accordingly (see Participants sub-section of the Methods).

We also now avoid the term “insomnia severity” to avoid confusion with the ISI, and only refer to the markers of objective sleep quality.

7. Be more specific when mentioning sleep stages: It should be clearer that expert staging is mostly meant in wordings such as "during wake". Consider that there are sleep stages as classified by the ML algorithm as well.

We thank the reviewer. Now we have clarified when we refer to expert stages. We have added the following sentence in the Results section (p. 6):

“In the following, any mention of sleep stages will refer to those scored by experts, while mentions to stage probabilities or intrusions refer to the probabilities obtained from the hypnodensities.”

Further specific comments:

8. "Increased intrusions in insomnia": "For this reason, N1 was excluded from the following analysis." And still, N1 is described to be included in all feature vectors (one of 5 or 6 sleep stages). Was it excluded or not?

N1 epochs as scored by experts were indeed excluded from our statistical analyses. However, to retain the integrity of PSG recording, we still considered it in the computation of HD, entropy and DKL. We have now clarified that in the Results section (p. 6):

“For this reason, N1 was excluded from the expert stages to be analyzed, but it was nevertheless considered in the computation of HD, entropy and DKL for completeness.”

Discussion:

9. Our algorithm is fairly simple and can be opened up": Instead of "open up", a commonly used term nowadays is "explainable", as in "explainable AI/ML".

Thank you. We have now changed the term into explainable, as suggested.

Data analysis:

10. "Intrusions and stability": The authors in several places describe their DKL measure as "stability" but it is actually *instability*, as it measures the mean differences between successive epochs. For example also on line 283: "The average intrusions and stability in wakefulness had the largest correlation with WASO and SE, with a negative correlation for the former and a positive for the latter" which would not be understandable because average intrusions and stability should be anticorrelated - but what the authors call "stability" is DKL, ie instability.

We have now changed the term from stability to instability in the Figures and text. However, note that the DKL measures the information loss when going from one probability distribution to another, which is different from the mean difference between successive epochs. DKL provides a more complete notion of the difference between epochs.

11. It is unclear, BTW, how "successive epochs" are defined for DKL. I assume that this is after filtering for sleep stage, meaning that both differences between actually successive epochs and differences between epochs of the same sleep stage separated by other stages are included. The latter pairs should be excluded.

We thank the reviewer for raising this important point. Indeed, we computed only intra-stage transitions, not inter-stage transitions. We have now made this point clearer both in the Results and Methods section.

Results (p. 8): *“we quantified the level of instability between two consecutive HDs within the same sleep stage by the DKL”*

Methods (p. 21): *“to measure the instability of an epoch with respect to the next one within the same sleep stage”*

12. Ethics: Wouldn't the GDPR be more relevant? Legally, local guidelines cannot deviate from it.

Since the data were collected in France and the research was performed in France, we are subject to regulations applicable in France. These regulations are of course in line with EU regulations (which include GDPR) but they can go beyond these regulations. The mention of French legal regulations is therefore more relevant.

Reviewer #2 (Remarks to the Author):

1. This paper uses machine learning to study hypodensities and entropy in insomnia individuals (INS). Its main aim is to quantify/qualify sleep state misperception (SSM). By comparing individuals (INS) presenting sleep difficulties to individuals presenting SSM and good sleepers, authors have set forward new investigation methods which contribute to our understanding of SSM. In fact, throughout their paper, results showed that SSM have sleep intrusion during wake in the night while INS have wake intrusions during sleep. Furthermore, it appears that these results indicate that while SSM individuals do not differ from GS in many respects, they do when the microstructure of sleep is considered. Authors conclude, with adequate emphasis that sleep, and its mismeasurement might be distributed along a continuum and that it is only by deciphering the EEG that one can justly estimate the contribution of stability and intrusions to sleep quality. This paper is one of the few I have been truly excited to read and review during my career. As has been observed in my own lab, it is by looking at the EEG and using fine tune methods that one can really quantify sleep/wake disruptions in INS. Insomnia disorder pertains to a large array of symptoms and encompasses the mere subjective complaints in addition to showing a wide range of heterogeneity within suffering individuals. This paper is very enlightening and set the way forward for using similar analyses with other groups of disorders and for, by example, attending to the contribution of one disease/ disorder or another. It also suggests that we should stop trying to classify individuals into categories which I find reassuring. Those seemingly on the 'verge' of being into one or the following categories is subjected to seemingly arbitrary cut-offs, and it should not be. Thanks for this brilliant work.

We are grateful for this positive feedback.

Here are few comments which should be addressed, if not, seriously considered.

2. It appears that not much allusion is made to the fact that INS often show good and bad nights (the disorder does specify 3 or more nights per week). By defining sleep quality as they did, do authors feels that their results would differ according to bad or good nights? If so, how could we explain those?

Thank you for this methodologically important comment. We took the approach to define SOSD based on individuals' subjective complaints and one night of PSG recordings. We agree that our

approach does not take into account the variability across nights, which is now mentioned explicitly in the Methods section (p. 18):

“SOSD denotes a disjunction between perceived and objective sleep patterns. This discrepancy can be established in various ways, for example by comparing PSG metrics and subjective reports on the same night. Here, we compared a general complaint of poor sleep (falling within the diagnostic criteria of Chronic Insomnia) and the PSG results obtained in one night of sleep recording. [...] This approach does not take into account the variability in PSG outcomes across nights but is well adapted to clinical settings where, typically, only one night of PSG is performed.”

Importantly, our approach relies on the data available to practitioners, who will have access to patients' subjective complaints and, sometimes, one night of PSG if such recording was administered. This is a real advantage of this study since it means our approach can be easily transferred to current clinical practices, helping practitioners make sense of a potentially surprising PSG outcome. However, it is true our approach is subject to inter-night variability and characterises more the PSG recording that was performed than the individual that was recorded. To acknowledge these limitations, we revised our Discussion as follows (p. 15):

“Another limitation is that the differences we observe between SOSD- and SOSD+ relate to a single night of polysomnography per subject which is compared to a subjective complaint of insomnia that covers a previous and longer period of time. This is in line with the data that can be available to practitioners (a subjective and chronic complaint of insomnia associated with a punctual assessment of sleep through a PSG). However, this approach does not account for the night-to-night variability in PSG recordings, and phenomena such as the first-night or reverse first-night effects. It is thus possible that the SOSD+ and SOSD- labels could change from night to night at the individual level. Future studies could employ other approaches to characterise SOSD for a specific night by comparing PSG recordings with subjective assessment of the same night. The same analytic approach as presented here could be applied.”

3. Authors also suggest that results could be geared towards bettering treatment. They offer very poor alternatives or suggestions in that regards. They should be more explicit about what they suggest.

The current reference treatment for insomnia is cognitive-behavioral therapy for insomnia (CBTi), whose effectiveness is recognized regardless of the type of chronic insomnia. We have included this in the Discussion section (p. 14):

“Indeed, the current reference treatment for insomnia is cognitive-behavioral therapy for insomnia (CBTi), whose effectiveness is recognized regardless of the type of chronic insomnia. We propose that the notions of “intrusions”, “stability” and “mixing” of sleep states represent useful concepts to explain the correspondence between what is measured by the PSG and what is perceived by individuals, to enable certain patients to better understand the complexity of brain states during sleep. These notions could help practitioners explain to patients the discrepancy between a normal PSG outcome and their insomnia complaint. Prospective studies could also leverage this new approach to personalise treatments by investigating if metrics derived from PSG recordings can predict treatment outcomes.”

4. Any ideas on the neuronal mechanisms underlying sleep/wake mismeasurements besides the theory of local cortical vigilance states?

We outline additional possibilities for the mechanisms of SOSD: (1) circadian misalignment could favour stage mixing, (2) metabolic changes that can alter brain dynamics and wake/sleep regulation, and (3) disruption of noradrenergic modulation could also perturb sleep architecture.

These mechanisms are not mutually exclusive and are also compatible with the local sleep hypothesis already mentioned in the manuscript. We now dedicate a full paragraph to this Discussion (see the corresponding references in the main text):

“Previous findings on sleep and SODS could explain the increase in entropy and instability that we observed in the two insomnia groups (Figure 2). Indeed, an increase in faster oscillations is associated with the feeling of being awake while asleep (FAWS), as assessed directly when awakening individuals from sleep. In insomnia patients, this association between FASW and faster oscillations is present in both NREM and REM sleep. It is likely that such increases in faster oscillations within sleep epochs would lead to an increase in the entropy of hypnodensities similar to the one we report here. Such modulations of faster oscillations within sleep have also been associated with parasomnia, dreaming, and cognitive processes during sleep, which suggests links between local modulations of sleep and a broad range of sleep disruptions including insomnia with SODS. Local modulations of cortical dynamics could thus help close the gap between the objective and subjective assessment of sleep by stressing the importance of global and local dynamics within sleep. Our approach could help quantify these local changes beyond the standard global classification of sleep stages. These local dynamics could themselves be caused by a misalignment of sleep with the circadian phase, metabolic changes (e.g., higher metabolic activity during sleep in certain brain regions), or a disruption of the neuromodulation of sleep/wake transitions (e.g., a higher noradrenergic activity during sleep). These factors have been shown to impact sleep and provide interesting avenues of research to better understand and treat insomnia.”

5. Sleep quality is a difficult concept to grasp, define. For one it will mainly be the ‘type’ of sleep difficulties while for the other, the number of arousals during the night. What would authors suggest being different between those different types?

Sleep quality is indeed multidimensional, which is why we took different indexes of sleep quality: sleep onset latency, sleep efficiency and total sleep time. There could be many more ways to estimate sleep quality both at the subjective and objective level but this reflection and better characterisation of sleep quality represent a research project on its own and we think it merits a dedicated article.

6. The reviewer’s understanding is that the selected EEG was ‘very clean’. What does it mean? Does it mean that stage transitions with movements or short awakenings were discarded? It is understood that AASM criteria were used for the scoring. How many epochs were ‘un-scorable’ and how many were rejected?

We would like to clarify that we did not use the term “very clean” in our manuscript, and it appears that this may have been a misunderstanding on the reviewer’s part. All epochs from all PSG recordings were manually scored according to AASM criteria to define sleep stages. No epochs were excluded from the manual annotations, and we did not remove any stage transitions or awakenings. 115 recordings with excessive artifacts leading to poor automatic sleep staging performance (<0.4) were discarded from the outset.

7. It appears that SSM were twice as subjected to cardiovascular diseases than INS. How could this influence the data?

We believe this is a confusion on the side of the Reviewer. In Table 1 of the originally submitted manuscript, we have 13.8% of the SODS- population with cardiovascular diseases and 5.2% for SODS+, so the opposite effect. We now show that this difference is non-significant in the Supplementary Table 1 of the revised version of the manuscript.

Reviewer #3 (Remarks to the Author):

1. Thank you for the opportunity to review this interesting paper approaching the phenomenon of SSM in Insomnia Disorder using two measures of intrusions and stability of sleep through advanced machine learning techniques. The rationale and the aim of the study are paramount in the study of sleep perception in insomnia patients, and the methodologies applied could provide new and important information on the electrophysiological processes underlying SSM. However, some methodological choices severely limit the quality of the paper.

We appreciate the reviewer's comments. We sought to address the Reviewer's methodological concerns, in order to clarify and improve our manuscript.

2. The main methodological limitation of the study is the definition of the SSM groups as subjects who complained of insomnia symptoms during the clinical evaluation but did not exhibit PSG abnormalities during the recording night. However, this definition does not account for the first-night effect and, especially, for the night-to-night variability in insomnia symptoms (Buysse et al., 2010; Newell et al., 2012).

We thank the reviewer for raising this issue. We have addressed this issue in response #2 for Reviewer #2 and modified the Methods and Discussion sections:

“SOSD denotes a disjunction between perceived and objective sleep patterns. This discrepancy can be established by comparing PSG metrics and subjective reports on the same night. Here, we compared a general complaint of poor sleep (falling within the diagnostic criteria of Chronic Insomnia) and the PSG results obtained in one night of sleep recording. [...] This approach does not take into account the variability in PSG outcomes across nights but is well adapted to clinical settings where, typically, only one night of PSG is performed.”

We took the approach to define SOSD based on individuals' subjective complaints and one night of PSG recordings (see General Comment at the top of this rebuttal letter). This approach has some limitations but corresponds to the data available to practitioners, who will have access to patients' subjective complaints and, sometimes, one night of PSG recordings if such recording was administered. We think this is a real advantage of this study since this means our approach can be easily transferred to current clinical practices. It can help practitioners make sense of a potentially surprising PSG outcome. However, our approach is indeed subject to inter-night variability and characterises more the PSG recording that was performed than the individual that was recorded. It thus leaves some questions open and we revised our Discussion accordingly:

“Another limitation is that the differences we observe between SOSD- and SOSD+ relate to a single night of polysomnography per subject which is compared to a subjective complaint of insomnia that covers a previous and longer period of time. This is in line with the data that can be available to practitioners (a subjective and chronic complaint of insomnia associated with a punctual assessment of sleep through a PSG). However, this approach does not account for the night-to-night variability in PSG recordings, first-night or reverse first-night effects and we can expect that the SOSD+ and SOSD- labels could change from night to night at the individual level.

However, our goal was to characterise the phenomenon of SODS in general, not to draw detailed inferences on specific individuals. Future studies could employ other approaches to characterise SODS as an individual trait or to examine the neural correlates of SODS for a specific night. The analytic approach presented here could represent an interesting framework for such an endeavor.”

3. Moreover, how long has it been since the patient complained of symptoms of insomnia during the clinical visit and when PSG was performed?

Following the typical clinical protocols of the Hotel Dieu Hospital, there was an average 6 week delay, during which there is no expectation that the diagnosis will change, as they come after a much longer period complaining about insomnia. This delay is due to administrative issues.

4. The degree and definition of the SSM are very difficult to establish, and the standard way to measure the SSM degree is to collect the subjective sleep times perceived by the individual the morning following the PSG recording, a few minutes after the final awakening to ensure an accurate and unbiased memory of the perception of night sleep (Castelnovo et al., 2019). In my opinion, the way used by the authors to discriminate insomnia patients with and without SSM is not appropriate from a methodological standpoint.

Thank you for this methodologically important comment. We clarified our approach in the Methods section (p. 18):

“SODS denotes a disjunction between perceived and objective sleep patterns. This discrepancy can be established in various ways, for example by comparing PSG metrics and subjective reports on the same night. Here, we compared a general complaint of poor sleep (falling within the diagnostic criteria of Chronic Insomnia) and the PSG results obtained in one night of sleep recording.”

We argue that SODS/SSM can be assessed in many ways and the appropriateness of a given methodological approach depends on the goals of each study. We agree that a force-awakening approach in which participants are asked to provide their subjective impression of sleep (or wakefulness) is optimal for identifying mechanisms of the “feeling of being awake while asleep”, and we discuss the papers employing this approach. However, it is less suited for clinical practice. For example, it disrupts sleep and could exacerbate SODS if participants are expected to be awakened (on-call sleep). We therefore opted for an approach that can be directly applied to clinical settings by estimating the SODS between a chronic complaint of insomnia and the objective sleep metrics derived from one night of PSG. Of course, this approach also has limitations that we detail in the Discussion (p. 15):

“Another limitation is that the differences we observe between SODS- and SODS+ relate to a single night of polysomnography per subject which is compared to a subjective complaint of insomnia that covers a previous and longer period of time. This is in line with the data that can be available to practitioners (a subjective and chronic complaint of insomnia associated with a punctual assessment of sleep through a PSG). However, this approach does not account for the night-to-night variability in PSG recordings, first-night or reverse first-night effects and we can expect that the SODS+ and SODS- labels could change from night to night at the individual level. However, our goal was to characterise the phenomenon of SODS in general, not to draw detailed inferences on specific individuals. Future studies could employ other approaches to characterise SODS as an individual trait or to examine the neural correlates of SODS for a specific night. The analytic approach presented here could represent an interesting framework for such an endeavor.”

5. Another point regarding the insomnia disorder group, is the definition of inclusion/exclusion criteria. Has the exclusion of other sleep disorders been established? How? explain in the text.

The PSG itself and the clinical data available for each patient enabled us to exclude patients with obstructive sleep apnea (OSA) or periodic limb movement disorder (PLMD). We added the following details to the Methods (p. 17):

“Patients were screened for obstructive sleep apnea (OSA) or periodic limb movement disorder (PLMD) by analysing PSG recordings and following existing guidelines (ICSD-3). All individuals with OSA and PLS were excluded from our analyses.”

6. Moreover, as a minor point, in the Objective Insomnia paragraph in the methods section authors reported that delineated three subtypes of insomnia based on the objective PSG indices of SOL (i.e., Sleep onset insomnia subtype), WASO (i.e., Sleep maintenance insomnia subtype), and the morning awakening (i.e., early morning awakening insomnia subtype). However, the analyses do not account for this subtyping. What was the purpose of this distinction?

We apologize if this was unclear. We did not perform such a subtyping of the SOSD+ group and we modified the Methods section to avoid this confusion (p. 18):

“We leveraged PSG recording to identify patients with significant objective disruption of their sleep as measured by a single-night of PSG: sleep onset insomnia (SOL > 30 minutes), OR sleep maintenance insomnia (WASO > 30 minutes), OR early morning awakening insomnia (natural awakening > 1 hour before desired wake time). Subjects meeting at least one of these criteria were categorized as “SOSD-.”

7. As a second methodological limitation, the study is based on the analysis of sleep intrusion during wake epochs and, vice versa, of wake intrusion during sleep epochs. However, the methodological approach used by the authors does not allow to distinguish which type of sleep intrusions are present during wake, and in which specific sleep stages the waking intrusion occurs. In other words, from a physiological and processual point of view, it is totally different if a wake intrusion occurs during an epoch of N1 or N3 sleep, and, on the other hand, it will be very different if, during a wake epoch, there will be an intrusion of REM sleep or SWS. Referring generally to wake or sleep intrusions without a characterization of these physiological states is, in my opinion, not informative in the process of SSM and is sometimes misleading. For example, you defined levels of entropy as “high HD entropy indicates a high level of intrusion (proximity to a uniform distribution), high entropy during wakefulness implies sleep intrusions, while high entropy during sleep implies wakefulness or intrusions from other sleep stages”. However, when you reported findings in the three groups, results of larger entropy during N2 and N3 are interpreted to “an increase in wake probabilities”. But are we sure that these intrusions are of wake and not of other sleep stages if the analysis is aspecific to the type of intrusions that occur?

We appreciate the reviewer’s comment and we indeed agree with this perspective. In fact, this is exactly what Figure 2A shows, but in the originally submitted version there was a typo and we cited Figure 1A instead of Figure 2A. Also, in the originally submitted version we address exactly this point in the following paragraph:

“When examining group differences, we found a significant reduction (Bonferroni corrected Wilcoxon-rank sum $p < 0.001$) of the wake prediction probability during expert-scored wakefulness for SOSD- (0.70 ± 0.16) and even less for SOSD+ (0.49 ± 0.17), compared to GS (0.75 ± 0.18) (see Supplementary Table 2 for HD details). While SOSD- exhibited a significantly larger probability of wakefulness during sleep (NREM+REM), SOSD+ did not (GS: 0.091 ± 0.14 ; SOSD-:

0.12 ± 0.12 ; $SOSD+$: 0.067 ± 0.080); *Bonferroni corrected Wilcoxon-rank sum GS vs SOSD-* $p < 0.001$, *GS vs SOSD+* $p > 0.01$). *In fact, SOSD- exhibited larger wake intrusions (defined here as the wake probability in sleep epochs) than SOSD+ in all sleep stages. “*

Despite the richness of information encoded in the identity of the different sleep intrusions during sleep (e.g. N3 into N2), the main observed effect was in wakefulness. However, we agree with the Reviewer that it would be interesting to dive into finer detail in the subtypes of awakenings. Yet, this approach could prove complex and we preferred the simplicity of deriving just two but highly informative metrics: entropy and dKL.

8. The same lack of characterization is detrimental in interpreting the results of sleep stability. Indeed, it is very different if a period of N3 is followed by epochs of N2, REM sleep, or wake. Without this type of distinction, these results are open to speculation not supported by the findings.

We have clarified this issue in the response to Reviewer #1's comment #11. The instability was measured only for intra-stage epochs and it is now better clarified in the manuscript (p. 21): *“to measure the instability of an epoch with respect to the next one within the same sleep stage, we used the DKL to quantify the difference between the two respective HD”*.

9. In these terms, another point is the interpretation of results about stability. Authors report: “... only SSM shows an increased instability in wakefulness compared to GS. Again, these results are consistent with the condition of SSM as a disruption of intra-sleep wakefulness rather than sleep itself.”. My question is: is it permissible to interpret the result of wake periods interrupted by sleep as disruption of intra-sleep wakefulness? In my opinion, this result may be interpreted as a wake-sleep pattern fragmented in SSM patients, in terms of difficulty in reaching stable sleep.

Considering that instability was measured only for intra-stage epochs, the increased instability in wakefulness is not driven by fragmentation, as the transition towards other stages were not considered.

10. Moreover, it could be interesting and informative to report at what time of night intrusions and stage shifting occur (i.e., it is very different if an intrusion of N3 sleep during wake occurs at the beginning or at the end of the night, or if the wake instability occurs at the sleep onset or in the middle of the night). Analyses of sleep cycles may be informative.

We appreciate this comment and have conducted an ANOVA analysis to test the interaction between group differences and each third of the night (e.g. splitting the night in 3 consecutive segments of approximately the same length). We found no significant interaction between the group differences and the third of the night, so we did not perform further analysis. For completeness, we attach the figures.

We agree it could be interesting to analyse each sleep cycle. However, the number of sleep cycles vary between individuals and the identification of sleep cycles is not trivial, especially in the SODS-group.

Other limitation points regard the methods used to record sleep and extract PSG indices.

- As a retrospective study, the number of EEG derivations varied across devices used to record sleep. Please report, also in supplementary materials, the different EEG montages used in the study.

We report now this more detailed information in the Methods section (p. 19):

“The NOX-A1, MORPHEUS, and SOMNOLOGICA systems included at least two EEG electrodes from C3, C4, F3, F4, O1, and O2, all referenced to their contralateral mastoid. The ACTIWAVE system used at least two of the EEG electrodes O1, O2, C3, F3, and FP1, also referenced to their contralateral mastoid.”

- In the Preprocessing paragraph is not reported if an artifact rejection was carried out for the power spectral analyses. Please, specify whether and how (i.e., manual or automatic) it was carried out.

We appreciate this comment and have added a few sentences specifying this preprocessing step (p. 20):

“PSDs were computed only for epochs that were scored as W, N1, N2, N3 or REM. Noisy epochs were usually not included because they were not scored as one of these 5 states if the signal quality did not allow for accurate scoring. We did not perform further data cleaning.”

- Regarding TST and duration of each sleep stage, absolute (i.e., in minutes) or percentage indices were used?

We have now included the unit (minutes) in the figures (3 and 4) and the text.

14. In the Spectral analysis paragraph, it is not reported if the power is relative or absolute and if the spectral analyses were carried out for single bands, and for single sleep stages or NREM/REM sleep. Please, specify these issues in the text.

We have now specified this information in the Methods section (p. 20):

“Then, for each valid epoch, we extracted the normalized the power in 6 canonical bands (delta: 0.5-4 Hz, theta: 4-8 Hz, alpha: 8-12 Hz, sigma: 12-16 Hz, beta: 16-30 Hz and gamma: 30-40 Hz). Normalization was performed by dividing the PSD values by the sum of the whole PSD.”

15. The text does not present a description of the sleep macrostructure in the three groups examined. Please report PSG macrostructure indices and the statistical between-group differences, also through a table.

We apologize for not adding this in the first version of our manuscript. This information is now included with its respective statistical test in Supplementary Table 6, 7, 8 and 9. We now mention these tables in the main text, and in the Method's section.

16. Besides the mentioned methodological limitations, the paper, in my opinion, presents other conceptual, interpretation, and presentation problems. First of all, the way in which results of “Increased intrusions in insomnia: sleep intrusion in SSM and wake intrusion in INS” are reported makes them difficult to understand. Especially the first part of the description of the results is not accompanied by a clear image to help to understand the findings. I guess that the results described refer to Figure 2 A; however, this is never cited in the text. Please, provide a better explanation of these findings and pay attention to the figure.

We thank the reviewer for picking up this issue. Indeed, we made a typo and cited Figure 1A, instead of figure 2A. We have now corrected this hoping this will make the presentation of the results clearer.

17. In the “Sleep intrusions and wake stability predict diagnosis and sleep quality” paragraph authors report: “The average intrusions and stability in wakefulness had the largest correlation with WASO and SE (Figure 4C), with a negative correlation for the former and a positive for the latter. Accordingly, our results suggest that the level of intrusions and stability in wakefulness can not only discriminate between sleep groups but also predict the severity of insomnia.” In my opinion, this interpretation is tautological. Indeed, the negative correlation between the average intrusion and stability in wakefulness and the amount of WASO is an effect of the wake stability, on the other hand the positive correlation with the SE is the other side of the coin due to the increase of sleep.

It is important to note that these metrics are not directly related because the entropy and instability of wakefulness are metrics that describe the quality and not the quantity of wake epochs within a given PSG recording. As such, wake instability does not automatically reflect the amount of wakefulness (and therefore the WASO value). This is well illustrated in Figure 2C that shows that SOSD+ patients have higher dKL values for wakefulness compared to GS despite a similar number of wake epochs. Furthermore, since dKL was computed between epochs with the same stage, dKL is not trivially related to the number of stage transition (and therefore awakenings). We therefore argue that these correlations could be capturing something different –particularly something about brain activity– rather than being directly related to WASO or SE.

18. Moreover, why the authors interpreted these results as predictors of the severity of insomnia?

We agree with the Reviewer #2 (and Reviewer #1) that using the term “insomnia severity” was inadequate here and we now use the term “sleep quality” instead.

19. It would be useful to include correlation coefficients when reporting correlational results in the text. The way in which Figures 4C, 5 A, B, C, D, and F are shown does not allow accurate reading of the results. About Figures 4C, what are the significant correlations (report p plus rho)?

We thank the suggestion. In Figure 4C we have identified the significant correlation by coloring them, while the non-significant correlation is depicted with grayscale in the correlation matrix. In Figure 5 we have clarified in the caption that the values are the population average of the correlations. So, for each subject we computed its respective correlations between e.g. entropy and delta band, and then we took the average across the whole population.

20. Why do authors sometimes use the two decimal places after the comma and in other cases 3, 4 or 5? It would be necessary to be uniform.

We have uniformed this.

21. Considering Figures 5A, B, C, D, and F, why did the authors graph correlational results through bar plots and not scatter plots? Please provide scatter plots of correlational results, so that the distribution of the single points of the correlation can be observed.

As replied in point 19, the correlations are the population average correlation, which has been made clearer in the revised version.

Minor revisions

• Introduction session:

22. When you report the reference number 16 there is a double bracket.

Thanks for the careful reading. It is now corrected.

23. You have rightly reported that “SSM is present in all sleep disorders” without mentioning that, although to a lesser degree, this is a phenomenon also recognized in good sleepers. I recommend a brief mention.

We have included this suggestion (p. 2):

“SOSD+ is present in all sleep disorders and, although to a lesser degree, this is a phenomenon also recognized in good sleepers”

24. In the period “Based on this relationship, the current consensus is that the detection of certain...” you defined the EEG acronyms. However, had already defined a few lines above.

Thanks, it is now corrected.

25. When you addressed the topic of the current lack of characterization and knowledge of the mechanisms underlying SSM and sleep perception due to the limitations of PSG analysis, I suggest citing a recent work where a topographical analysis was carried out: “Fasiello, E., Gorgoni, M., Galbiati, A., Sforza, M., Berra, F., Scarpelli, S., ... & De Gennaro, L. (2024). Decreased Delta/Beta ratio index as the sleep state-independent electrophysiological

signature of sleep state misperception in Insomnia disorder: A focus on the sleep onset and the whole night. *NeuroImage*, 298, 120782.”

Thanks for the suggestion. Indeed, we became aware of this paper after the submission of the original version. We have included a mention to this work in the Introduction and Discussion:

Introduction (p. 3):

“A recent examination of the topography of PSG has associated diffuse cortical hyperactivations with SODS, identifying a decrease in the delta/beta ratio as a major correlate³⁴.”

Discussion (p. 13):

“Despite the lack of spatial resolution of our work, our results are compatible with the recent finding of diffuse cortical hyperarousal as a marker for SODS³⁴, which could be also interpreted as sleep/wake mixing.”

- Results session

26. NREM and REM acronyms are never defined in full in the text. Please, define.

Thanks. They have been defined now in the introduction.

27. Figure 2 is titled “Intrusions and stability on insomnia with and without SSM.” However, the figure illustrates also findings in GS. Please, specify in the figure description.

Thanks, we have also included GS in the figure description.

28. The Figure 2A is not cited in the text.

This has been corrected.

29. In the “Sleep intrusions and wake stability predict diagnosis and sleep quality” paragraph the acronyms of TST, SOL, WASO, and SE are defined. However, these acronyms were yet defined in the text. Please uniform.

Thanks, we have made this uniform.

- Discussion session

30. Uniform the sleep state misperception term in the first period of the paragraph with the acronym SSM as in the rest part of the text.

Thank you. As explained above, we changed our nomenclature throughout

31. Authors defined the PSG acronym in the first period of the paragraph, but it is yet defined in the text. Please, uniform.

Thanks, we have made this uniform.

- Materials and methods session

32. In the Participants paragraph authors defined the PSG acronym, but it is yet defined in the text. Please, uniform.

Thanks, we have made this uniform.

33. Table 1: provide indices of statistical differences between the three groups, in addition to descriptive results

Thanks for this suggestion. We have now included a Supplementary Table 1 with the group comparisons.

34. The description of the recruitment procedure as a single paragraph does not help the reading. Please report the specific recruitment procedures for the single groups in the relative paragraphs (i.e., Insomnia without SOSD, Insomnia with SOSD, and Control Group).

Thank you for your suggestion regarding the presentation of the recruitment procedure. We have removed the overarching recruitment procedure paragraph and incorporated specific recruitment information directly into each respective section.

35. Please, cite in the text original studies in which the good sleepers were recruited (i.e., VIVACOM).

We have now included the list of previous studies where the GS were recruited in the subsection of GS recruitment..

36. In the Control Group paragraph recruited individuals were defined as “patients”. Please correct this. Moreover, the authors report that GS underwent rigorous screening to exclude confounding factors such as comorbidities. Which types of comorbidities were excluded?

Good sleepers were selected on the basis of having been free of sleep disorders for at least one year. In addition, subjects with chronic unbalanced pathologies, psychotropic treatments or those that could affect sleep were excluded from the protocols. Shift and night workers, people exposed to jet-lag or irregular sleep schedules (based on a 2-week sleep diary) were also excluded from these VIFASOM studies. In the Methods section we have included (p. 19):

“In addition, subjects with chronic unbalanced pathologies, psychotropic treatments or those that could affect sleep were excluded from the protocols. Shift and night workers, people exposed to jet-lag or irregular sleep schedules (based on a 2-week sleep diary) were also excluded”

Below are our response to the reviewers.

Reviewers' comments:

Reviewer #1 (Remarks to the Author):

The authors have responded well to the extensive review, clearly enhancing the manuscript. I have no further suggestions.

We thank Reviewer #1 for their critical feedback.

Reviewer #2 (Remarks to the Author):

In my opinion, the authors have adequately address all reviewers comments. Congratulations on a great addition to the field of insomnia.

Likewise, we wish to thank Reviewer #2 for their positive assessment and their contribution to our paper.

Reviewer #3 (Remarks to the Author):

Thank you to the authors for responding promptly to all my comments. However, sadly, all the reviews made by the authors in accordance with the requests of the reviewers do not fill the enormous methodological gap of this work. Specifically, regarding question number 3, "How long has it been since the patient complained of symptoms of insomnia during the clinical visit and when PSG was performed?" authors have reported that between the subjective report of sleep disturbance and the objective sleep recording have passed an average of 6 weeks. This approach is inadmissible to measure the subjective-objective sleep discrepancy; for example, it does not consider that in perspective of the night-to-night variability, how many of the tested patients would have reported sleeping well at night from PSG recording. All work is presented with a focus on a phenomenon (i.e., SSM, or the subjective-objective sleep discrepancy) that has been evaluated and conceptualized in an inaccurate way.

We thank Reviewer #3 for the time they dedicated to this manuscript. The primary concern raised by Reviewer #3 is that we cannot accurately estimate the subjective-objective sleep discrepancy (SOSD) because we did not collect subjective reports from patients on the night of their polysomnography (PSG) recordings. Instead, we compared patients with similar chronic insomnia complaints (all patients were diagnosed with insomnia based on their subjective complaints) based on their PSG outcomes (some patients show a normal PSG and some an abnormal one). We strongly disagree with Reviewer #3's characterization of this approach as "inadmissible."

Historically, SODS—or “paradoxical insomnia”—has been defined exactly as we did. For example, the 1979 Sleep Disorders Classification defines SODS as “a convincing and honest complaint of insomnia—made by an individual lacking apparent psychopathology—that is at variance with laboratory evidence of normal sleep length, architecture, and physiology.” According to this definition, paradoxical insomnia was identified at that time based on chronic insomnia complaints contradicting PSG findings. Importantly, paradoxical insomnia was not defined by comparing subjective assessments of sleep on the same night as the PSG. Our study follows exactly this definition of paradoxical insomnia (we preferred the more generic term of SODS in our manuscript).

While we acknowledge that later studies have sought to elucidate the **mechanisms** underlying SODS by comparing subjective and objective sleep assessments on the same night, this approach is better suited for mechanistic investigations rather than clinical practice. Our approach, though less suited for identifying specific mechanisms, offers findings that are more readily applicable in clinical settings. Thus, it is not accurate to label our approach as “inadmissible” when it aligns with the original definition of SODS and serves an important clinical purpose. We now stress this important distinction between the identification of biomarkers and mechanisms of SODS (p. 15):

“Our stance also corresponds to the classical definition of SODS in clinical practice (e.g., the definition of “paradoxical insomnia”). While this approach is well-suited for clinical practice and the identification of biomarkers of SODS, it is limited in its ability to uncover the underlying neural mechanisms of SODS. To address the question of mechanisms more specifically, obtaining subjective assessments of sleep, particularly following spontaneous or induced awakenings, is preferable.”

Regarding Reviewer #3’s additional criticism, we respectfully disagree with their assertion of an “enormous methodological gap.” We have explicitly stated the limitations of our approach in the revised manuscript, and these limitations were acknowledged by the other reviewers. Importantly, we are not making mechanistic claims about SODS but rather focusing on clinically relevant biomarkers.

Finally, we are at an impasse concerning Reviewer #3’s dissatisfaction with our response to their comment: “How long has it been since the patient complained of symptoms of insomnia during the clinical visit and when PSG was performed?” Unfortunately, we do not have this information. Given these constraints, we are unable to address this concern further.

In conclusion, our manuscript employs a definition of SODS that has been foundational in the field since its inception. While we acknowledge that this definition has limitations in elucidating neural mechanisms, it offers a unique perspective for identifying clinically relevant biomarkers of SODS and insomnia. We firmly believe that our approach is both methodologically sound and scientifically significant. We believe we have thoroughly explored all possible avenues to address Reviewer #3’s concerns.